# GC-Bench: An Open and Unified Benchmark for Graph Condensation

**Qingyun Sun**[1,*] **Ziying Chen**[1,*] **Beining Yang**[2], **Cheng Ji**[1], **Xingcheng Fu**[3]
**Sheng Zhou**[4], **Hao Peng**[1], **Jianxin Li**[1], **Philip S. Yu**[5]
[1]Beihang University, [2]University of Edinburgh, [3]Guangxi Normal University
[4]Zhejiang University, [5]University of Illinois, Chicago
{sunqy,chanztuying}@buaa.edu.cn

## Abstract

Graph condensation (GC) has recently garnered considerable attention due to its ability to reduce large-scale graph datasets while preserving their essential properties. The core concept of GC is to create a smaller, more manageable graph that retains the characteristics of the original graph. Despite the proliferation of graph condensation methods developed in recent years, there is no comprehensive evaluation and in-depth analysis, which creates a great obstacle to understanding the progress in this field. To fill this gap, we develop a comprehensive Graph Condensation Benchmark (GC-Bench) to analyze the performance of graph condensation in different scenarios systematically. Specifically, GC-Bench systematically investigates the characteristics of graph condensation in terms of the following dimensions: effectiveness, transferability, and complexity. We comprehensively evaluate 12 state-of-the-art graph condensation algorithms in node-level and graph-level tasks and analyze their performance in 12 diverse graph datasets. Further, we have developed an easy-to-use library for training and evaluating different GC methods to facilitate reproducible research. The GC-Bench library is available at https://github.com/RingBDStack/GC-Bench.

## 1 Introduction

Data are the driving force behind advancements in machine learning, especially with the advancement of large models. However, the rapidly increasing size of datasets presents challenges in management, storage, and transmission. It also makes model training more costly and time-consuming. This issue is particularly pronounced in the graph domain, where larger datasets mean more large-scale structures, making it challenging to train models in environments with limited resources. Compared to graph coarsening, which groups nodes into super nodes, and sparsification, which selects a subset of edges, graph condensation [39, 10] synthesizes a smaller, informative graph that retains enough data for models to perform comparably to using the full dataset.

**Graph condensation.** Graph condensation aims to learn a new small but informative graph. A general framework of GC are shown in Figure 1. Given a graph dataset $\mathbf{G}$, the goal of Graph condensation is to achieve comparable results on synthetic condensed graph dataset $\mathbf{G}'$ as training on the original $\mathbf{G}$. For Node-level condensation, the original dataset $\mathbf{G} = \{\mathcal{G}\} = \{\mathbf{X} \in \mathbb{R}^{N \times d}, \mathbf{A} \in \mathbb{R}^{N \times N}\}$ and the condensed dataset $\mathbf{G}' = \{\mathcal{G}'\} = \{\mathbf{X} \in \mathbb{R}^{N' \times d'}, \mathbf{A} \in \mathbb{R}^{N' \times N'}\}$, where $N' \ll N$. For Graph-level condensation, the original dataset $\mathbf{G} = \{\mathcal{G}_1, \mathcal{G}_2, \cdots, \mathcal{G}_n\}$ and the condensed dataset $\mathbf{G}' = \{\mathcal{G}'_1, \mathcal{G}'_2, \cdots, \mathcal{G}'_{n'}\}$, where $n' \ll n$. The condensation ratio $r$ can be calculated by condensed dataset size / whole dataset size.

---

*Equal contribution.

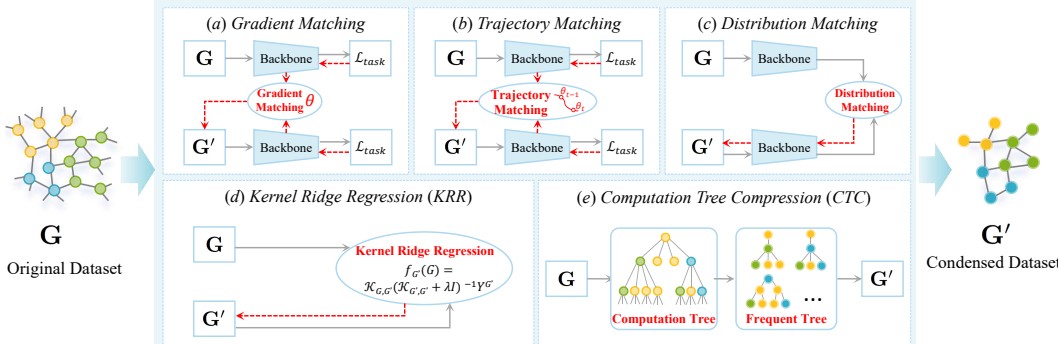

Figure 1: The GC methods can be broadly divided into two categories: The first category depends on the backbone model, refining the condensed graph by aligning it with the backbone's *gradients* (*a*), *trajectories* (*b*), and output *distributions* (*c*) trained on both original and condensed graphs. The second category, independent of the backbone, optimizes the graph by matching its distribution with that of the original graph data (*d*) or by identifying frequently co-occurring computation trees (*e*).

**Research Gap.** Although several studies aim to comprehensively discuss existing GC methods [39, 10], they either overlook graph-specific properties or lack systematic experimentation. This discrepancy highlights a significant gap in the literature, partly due to limitations in datasets and evaluation dimensions. A concurrent work [23] analyzed the performance of node-level GC methods but it included only a subset of representative methods, lacking an analysis of graph-level methods, a deep structural analysis, and an assessment of the generalizability of the methods. To bridge this gap, we introduce GC-Bench, an open and unified benchmark to systematically evaluate existing graph condensation methods focusing on the following aspects: ❶ **Effectiveness:** the progress in GC, and the impact of structure and initialization on GC; ❷ **Transferability:** the transferability of GC methods across backbone architectures and downstream tasks; ❸ **Efficiency:** the time and space efficiency of GC methods. The contributions of GC-Bench are as follows:

- *Comprehensive benchmark.* GC-Bench systematically integrated 12 representative and competitive GC methods on both node-level and graph-level by unified condensation and evaluation, giving multi-dimensional analysis in terms of effectiveness, transferability, and efficiency.

- *Key findings.* (1) Graph-level GC methods is still far from achieving the goal of lossless compression. A large condensation ratio does not necessarily lead to better performance with current methods. (2) GC methods can retain semantic information from the graph structure in a condensed graph, but there is still significant improvement room in preserving complex structural properties. (3) All condensed datasets struggle to perform well outside the specific tasks they were condensed, leading to limited applicability. (4) Backbone-dependent GC methods embed model-specific information in the condensed datasets, and popular graph transformers are not compatible with current GC methods as backbones. (5) The initialization mechanism affects both the performance and convergence according to the characteristics of the dataset and the GC method. (6) Most GC methods coupled with backbones and whole dataset training have poor time and space efficiency, contradicting the initial purpose of using GC for efficient training.

- *Open-sourced benchmark library and future directions.* GC-Bench is open-sourced and easy to extend to new methods and datasets, which can help identify directions for further exploration and facilitate future endeavors.

## 2 Overview of GC-Bench

We introduce **G**raph **C**ondensation **Bench**mark (**GC-Bench**) in terms of datasets (▷ Section 2.1), algorithms (▷ Section 2.2), research questions that guide our benchmark design (▷ Section 2.3) and the comparison with related benchmarks (▷ Section 2.4). The overview of GC-Bench is shown in Table 1. More details can be found in the Appendix provided in the **Supplementary Material**.

Table 1: An overview of GC-Bench

| Methods | |
|---|---|
| Traditional core-set methods | *Random, Herding* [36], *K-Center* [30] |
| Gradient matching | *GCond* [15], *DosCond* [14], *SGDD* [42] |
| Trajectory matching | *SFGC*[47], *GEOM* [45] |
| Distribution matching | *GCDM* [20], *DM* [24, 22] |
| Kernel Ridge Regression | *KiDD* [41] |
| Computation Tree Compression | *Mirage* [11] |
| **Datasets** | |
| Homogeneous datasets | *Cora* [16], *Citeseer* [16], *ogbn-arxiv* [13], *Flickr* [43], *Reddit* [12] |
| Heterogeneous datasets | *ACM* [46], *DBLP* [7] |
| Graph-level datasets | *NCI1* [32], *DD* [4], *ogbg-molbace* [13], *ogbg-molbbbp* [13], *ogbg-molhiv* [13] |
| **Downstream Tasks** | |
| Node-level task | Node classification, Link prediction, Anomaly detection |
| Graph-level task | Graph classification |
| **Evaluations** | |
| Effectiveness | Performance under different condensation ratios, Impact of structural properties, Impact of initialization mechanism |
| Transferability | Different downstream tasks, Different backbone model architectures |
| Efficiency | Time and memory consumption |

## 2.1 Benchmark Datasets

Regarding evaluation datasets, we adapt the 12 widely-used datasets in the current literature. The node level dataset include 5 homogeneous dataset (*Cora* [16], *Citeseer* [16], *ogbn-arxiv* [13], *Flickr* [43], *Reddit* [12]) and 2 heterogeneous datasets (*ACM* [46] and *DBLP* [7]). The graph-level dataset include *NCI1* [32], *DD* [4], *ogbg-molbace* [13], *ogbg-molbbbp* [13], *ogbg-molhiv* [13]. We leverage the public train, valid, and test split of these datasets. We report the dataset statistics in Appendix A.1.

## 2.2 Benchmark Algorithms

We selected 12 representative and competitive GC methods across 6 categories for evaluation. The main ideas of these methods are shown in Figure 1. The evaluated methods include: (1) traditional core-set methods including *Random*, *Herding* [36], *K-Center* [30], (2) gradient matching methods including *DosCond* [14], *GCond* [15] and *SGDD* [42], (3) trajectory matching methods including *SFGC* [47] and *GEOM* [45], (4) distribution matching methods including GCDM [20] and DM [22, 24], (5) Kernel Ridge Regression (KRR) based method *KiDD* [41], and (6) Computation Tree Compression (CTC) method *Mirage* [11]. The details of evaluated methods are in Appendix A.2.

## 2.3 Research Questions

We systematically design the GC-Bench to comprehensively evaluate the existing GC algorithms and inspire future research. In particular, we aim to investigate the following research questions.

**RQ1: How much progress has been made by existing GC methods?**

**Motivation and Experiment Design.** Previous GC methods always adopt different experimental settings, making it difficult to compare them fairly. Given the unified settings of GC-Bench, the first question is to revisit the progress made by existing GC methods and provide potential enhancement directions. A good GC method is expected to perform well consistently under different datasets and different condensation ratios. To answer this question, we evaluate GC methods' performance on 7 node-level datasets and 5 graph-level datasets with a broader range of condensation ratio $r$ than previous works. The results are shown in Sec. 3.1 and Appendix B.1.

**RQ2: How do the potential flaws of the structure affect the graph condensation performance?**

**Motivation and Experiment Design.** Most of the current GC methods borrow the idea of image condensation and overlook the specific non-IID properties of irregular graph data. The impact of structural properties in GC is still not thoroughly explored. On one hand, the structure itself possesses various characteristics such as homogeneity and heterogeneity, as well as homophily and heterophily. It remains unclear whether these properties should be preserved in GC and how to preserve them. On the other hand, some structure-free GC methods [47] suggest that the condensed dataset may not

need to explicitly preserve graph structure, and preserving structural information in the generated samples is sufficient. To answer this question, we evaluated GC methods on both homogeneous and heterogeneous as well as homophilic and heterophilic datasets to further explore the impact of structural properties. The results are shown in Sec. 3.2 and Appendix B.2.

**RQ3: Can the condensed graphs be transferred to different types of tasks?**

**Motivation and Experiment Design.** Most existing GC methods are primarily designed for node classification and graph classification tasks. However, there are numerous downstream tasks on graph data, such as link prediction and anomaly detection, which focus on different aspects of graph data. The transferability of GC across various graph tasks has yet to be thoroughly explored. To answer this question, we perform condensation guided by node classification and use the condensed dataset to train models for 3 classic downstream tasks: *link prediction*, *node clustering*, and *anomaly detection*. The evaluation results are shown in Sec. 3.3 and Appendix B.3.

**RQ4: How does the backbone model architecture used for condensation affect the performance?**

**Motivation and Experiment Design.** The backbone model plays an important role in extracting the critical features of the original dataset and guiding the optimization process of generating the condensed dataset. Most GC methods choose a specific graph neural network (GNN) as the backbone. The impact of the backbone model architecture and its transferability is under-explored. A high-quality condensed dataset is expected to be used for training models with not only the specific one used for condensation but also various architectures. To answer this question, we evaluate the transferability performance for 5 representative GNN models (*SGC* [37], *GCN* [16], *GraphSAGE* [12], *APPNP* [18], and *ChebyNet* [2]) and 2 non-GNN models (the popular *Graph Transformer* [31]) and simple *MLP*). We also investigated the performance variation of different backbones with the number of training steps. The evaluation results are shown in Sec. 3.4 and Appendix B.4.

**RQ5: How does the initialization mechanism affect the performance of graph condensation?**

**Motivation and Experiment Design.** The initialization mechanism of the condensed dataset is crucial for convergence and performance in image dataset condensation but remains unexplored for irregular graph data. To answer this question, we adopt 5 distinct initialization strategies (*Random Noise*, *Random Sample*, *Center*, *K-Center*, and *K-Means*) to evaluate their impact on condensation performance and converge speed. The results are shown in Sec. 3.5 and Appendix B.5.

**RQ6: How efficient are these GC methods in terms of time and space?**

**Motivation and Experiment Design.** As the GC methods aim to achieve comparable performance on the condensed dataset and the original dataset, they always rely on the training process on the original datasets. The efficiency and scalability of GC methods are overlooked by existing methods, which is crucial in practice since the original intent of GC is to reduce computation and storage costs for large graphs. To answer this question, we evaluate the time and memory consumption of these GC methods. Specifically, we record the overall time when achieving the best result, the peak CPU memory, and the peak GPU memory. The results are shown in Sec. 3.6 and Appendix B.6.

## 2.4 Discussion on Existing Benchmarks

To the best of our knowledge, GC-Bench is the first comprehensive benchmark for both node-level and graph-level graph condensation. There are a few image dataset condensation benchmark works for image classification task [1] and condensation adversarial robustness [38]. A recent work GCondenser [23] evaluates some node-level GC methods for node classification on homogeneous graphs with limited evaluation dimensions in terms of performance and time efficiency. Our GC-Bench analyzes more GC methods on a wider variety of datasets (both homogeneous and heterogeneous) and tasks (node classification, graph classification), encompassing both node-level and graph-level methods. In addition to performance and efficiency analysis, we further explore the transferability across different tasks (link prediction, node clustering, anomaly detection) and backbones. With GC-Bench covering more in-depth investigation over a wider scope, we believe it will provide valuable insights into existing works and future directions. A comprehensive comparison with GCondenser can be found in Appendix A.5.

Table 2: **Node classification accuracy (%)** (mean±std) across datasets with varying condensation ratios $r$. $\mathcal{H}$ denotes the homophily ratio [29]. The best results are shown in **bold** and the runner-ups are shown in underlined. Red color highlights entries that exceed the whole dataset performance.

| | | Traditional Core-set Methods | | | Distribution | | Gradient | | | Trajectory | | Whole Dataset |
|---|---|---|---|---|---|---|---|---|---|---|---|---|
| Dataset | Ratio($r$) | Random | Herding | K-Center | GCDM | DM | DosCond | GCond | SGDD | SFGC | GEOM | |
| *Cora* ($\mathcal{H}$=0.81) | 0.26% | $31.6_{\pm1.2}$ | $48.6_{\pm1.4}$ | $48.6_{\pm1.4}$ | $39.0_{\pm0.4}$ | $35.6_{\pm3.9}$ | $78.7_{\pm1.3}$ | $79.8_{\pm0.7}$ | $78.6_{\pm2.7}$ | $78.8_{\pm1.2}$ | $50.1_{\pm1.2}$ | $80.8_{\pm0.3}$ |
| | 0.52% | $47.1_{\pm1.7}$ | $56.0_{\pm0.6}$ | $44.7_{\pm2.7}$ | $42.3_{\pm0.7}$ | $38.5_{\pm4.4}$ | $81.1_{\pm0.1}$ | $80.2_{\pm0.7}$ | $81.2_{\pm0.5}$ | $79.2_{\pm1.8}$ | $70.5_{\pm0.9}$ | |
| | 1.30% | $62.3_{\pm1.0}$ | $69.9_{\pm0.8}$ | $62.0_{\pm1.3}$ | $63.4_{\pm0.1}$ | $63.2_{\pm0.5}$ | $81.2_{\pm0.4}$ | $80.2_{\pm2.2}$ | $81.6_{\pm0.5}$ | $79.4_{\pm0.7}$ | $78.5_{\pm1.9}$ | |
| | 2.60% | $72.4_{\pm0.5}$ | $74.2_{\pm0.6}$ | $73.5_{\pm0.7}$ | $73.4_{\pm0.2}$ | $72.6_{\pm0.7}$ | $79.6_{\pm0.2}$ | $80.5_{\pm0.3}$ | $81.8_{\pm0.2}$ | $81.7_{\pm0.0}$ | $76.1_{\pm4.7}$ | |
| | 3.90% | $74.6_{\pm0.6}$ | $76.0_{\pm0.6}$ | $77.4_{\pm0.3}$ | $76.4_{\pm0.5}$ | $75.3_{\pm0.1}$ | $78.9_{\pm0.1}$ | $79.8_{\pm0.4}$ | $82.1_{\pm0.8}$ | $81.5_{\pm0.5}$ | $78.3_{\pm2.0}$ | |
| | 5.20% | $77.1_{\pm0.6}$ | $76.7_{\pm0.4}$ | $76.5_{\pm0.6}$ | $78.4_{\pm0.0}$ | $77.9_{\pm0.2}$ | $79.7_{\pm0.6}$ | $77.8_{\pm3.6}$ | $82.0_{\pm1.4}$ | $81.4_{\pm0.0}$ | $80.4_{\pm0.8}$ | |
| *Reddit* ($\mathcal{H}$=0.78) | 0.05% | $45.2_{\pm1.7}$ | $54.6_{\pm2.4}$ | $50.7_{\pm2.2}$ | $61.1_{\pm0.4}$ | $60.6_{\pm0.7}$ | $50.1_{\pm1.8}$ | $84.4_{\pm2.6}$ | $83.6_{\pm1.7}$ | $87.2_{\pm2.1}$ | $73.6_{\pm1.2}$ | $94.0_{\pm0.2}$ |
| | 0.10% | $53.3_{\pm1.6}$ | $62.3_{\pm1.6}$ | $50.1_{\pm1.3}$ | $72.3_{\pm0.5}$ | $68.7_{\pm1.2}$ | $59.5_{\pm2.8}$ | $84.7_{\pm1.9}$ | $83.8_{\pm2.2}$ | $81.5_{\pm4.2}$ | $76.2_{\pm3.9}$ | |
| | 0.20% | $65.1_{\pm1.2}$ | $69.4_{\pm2.1}$ | $54.2_{\pm1.9}$ | $80.8_{\pm0.6}$ | $76.7_{\pm0.1}$ | $65.8_{\pm0.4}$ | $86.3_{\pm2.4}$ | $88.3_{\pm0.5}$ | $90.4_{\pm0.9}$ | $89.9_{\pm0.6}$ | |
| | 0.50% | $76.1_{\pm1.8}$ | $81.4_{\pm0.7}$ | $67.8_{\pm1.3}$ | $86.4_{\pm0.1}$ | $84.2_{\pm0.2}$ | $65.8_{\pm0.4}$ | $87.9_{\pm2.7}$ | $91.0_{\pm0.1}$ | $91.7_{\pm0.2}$ | $91.7_{\pm0.3}$ | |
| | 1.00% | $84.9_{\pm0.5}$ | $85.9_{\pm0.3}$ | $74.6_{\pm0.8}$ | OOM | $78.9_{\pm0.7}$ | $78.3_{\pm1.3}$ | $87.2_{\pm1.2}$ | $85.9_{\pm0.9}$ | $91.8_{\pm0.2}$ | OOM | |
| | 5.00% | $92.3_{\pm0.1}$ | $91.9_{\pm0.1}$ | $88.4_{\pm0.3}$ | OOM | OOM | OOM | OOM | OOM | $91.9_{\pm0.7}$ | OOM | |
| Citeseer ($\mathcal{H}$=0.74) | 0.18% | $39.3_{\pm1.9}$ | $36.6_{\pm2.6}$ | $36.6_{\pm2.6}$ | $30.6_{\pm2.5}$ | $32.1_{\pm1.5}$ | $71.8_{\pm0.1}$ | $71.3_{\pm0.6}$ | $71.7_{\pm3.2}$ | $70.7_{\pm1.1}$ | $63.7_{\pm0.4}$ | $71.7_{\pm0.0}$ |
| | 0.36% | $44.1_{\pm2.0}$ | $41.5_{\pm2.6}$ | $38.7_{\pm3.2}$ | $36.2_{\pm2.0}$ | $43.3_{\pm1.3}$ | $73.1_{\pm0.1}$ | $70.4_{\pm1.8}$ | $73.3_{\pm3.3}$ | $70.9_{\pm1.6}$ | $69.8_{\pm0.1}$ | |
| | 0.90% | $49.3_{\pm1.0}$ | $56.5_{\pm1.3}$ | $51.4_{\pm1.4}$ | $52.6_{\pm1.1}$ | $53.7_{\pm0.3}$ | $73.0_{\pm0.2}$ | $70.7_{\pm0.9}$ | $72.8_{\pm0.6}$ | $70.7_{\pm0.5}$ | $70.5_{\pm0.3}$ | |
| | 1.80% | $57.4_{\pm0.7}$ | $68.4_{\pm0.8}$ | $61.4_{\pm1.2}$ | $64.7_{\pm0.6}$ | $66.4_{\pm0.2}$ | $71.6_{\pm0.6}$ | $70.2_{\pm1.4}$ | $73.6_{\pm0.6}$ | $70.4_{\pm1.2}$ | $67.5_{\pm0.6}$ | |
| | 2.70% | $66.2_{\pm1.0}$ | $68.1_{\pm0.8}$ | $67.5_{\pm0.8}$ | $69.2_{\pm1.6}$ | $68.5_{\pm0.0}$ | $71.7_{\pm1.0}$ | $67.7_{\pm0.2}$ | $72.1_{\pm0.5}$ | $71.0_{\pm1.0}$ | $70.9_{\pm0.5}$ | |
| | 3.60% | $69.2_{\pm0.3}$ | $69.1_{\pm0.7}$ | $69.3_{\pm0.7}$ | $69.3_{\pm0.4}$ | $70.4_{\pm0.4}$ | $72.4_{\pm0.3}$ | $68.4_{\pm1.6}$ | $73.3_{\pm1.2}$ | $70.6_{\pm0.1}$ | $70.7_{\pm0.1}$ | |
| ogbn-arxiv ($\mathcal{H}$=0.65) | 0.05% | $10.7_{\pm0.9}$ | $32.7_{\pm0.9}$ | $36.9_{\pm1.2}$ | $51.1_{\pm0.2}$ | $40.4_{\pm3.0}$ | $59.5_{\pm1.0}$ | $60.0_{\pm0.0}$ | $59.7_{\pm0.2}$ | $68.2_{\pm2.3}$ | $64.5_{\pm0.9}$ | $71.5_{\pm0.0}$ |
| | 0.10% | $36.5_{\pm1.1}$ | $41.6_{\pm1.0}$ | $40.3_{\pm0.9}$ | $55.9_{\pm1.5}$ | $47.7_{\pm2.0}$ | $60.9_{\pm0.8}$ | $59.5_{\pm1.1}$ | $56.1_{\pm3.4}$ | $69.1_{\pm0.7}$ | $64.6_{\pm1.4}$ | |
| | 0.20% | $43.2_{\pm0.8}$ | $48.9_{\pm0.8}$ | $42.7_{\pm1.0}$ | $59.3_{\pm0.8}$ | $47.1_{\pm0.7}$ | $62.2_{\pm0.4}$ | $61.4_{\pm0.7}$ | $60.9_{\pm0.9}$ | $69.0_{\pm0.4}$ | $64.3_{\pm0.0}$ | |
| | 0.50% | $48.3_{\pm0.4}$ | $51.4_{\pm0.4}$ | $48.3_{\pm0.5}$ | $61.0_{\pm0.1}$ | $57.7_{\pm0.2}$ | $62.1_{\pm0.3}$ | $63.0_{\pm0.9}$ | $63.2_{\pm0.2}$ | $67.1_{\pm0.0}$ | $67.2_{\pm0.0}$ | |
| | 1.00% | $51.2_{\pm0.4}$ | $52.0_{\pm0.4}$ | $50.3_{\pm0.5}$ | $61.0_{\pm0.0}$ | $59.8_{\pm0.2}$ | $61.8_{\pm1.3}$ | $62.8_{\pm0.0}$ | $62.0_{\pm2.8}$ | $68.4_{\pm1.3}$ | $69.1_{\pm0.0}$ | |
| | 5.00% | $57.2_{\pm0.2}$ | $56.1_{\pm0.2}$ | $56.5_{\pm0.4}$ | OOM | OOM | OOM | OOM | OOM | $69.7_{\pm0.1}$ | $70.6_{\pm0.5}$ | |
| *Flickr* ($\mathcal{H}$=0.24) | 0.05% | $40.5_{\pm0.4}$ | $41.8_{\pm0.6}$ | $43.9_{\pm0.7}$ | $45.8_{\pm0.4}$ | $44.7_{\pm0.5}$ | $40.7_{\pm1.1}$ | $44.2_{\pm2.3}$ | $46.6_{\pm0.1}$ | $44.3_{\pm0.4}$ | $44.8_{\pm0.8}$ | $46.8_{\pm0.2}$ |
| | 0.10% | $42.0_{\pm0.6}$ | $41.4_{\pm0.6}$ | $41.4_{\pm0.8}$ | $47.5_{\pm0.2}$ | $45.4_{\pm0.1}$ | $43.2_{\pm0.0}$ | $44.2_{\pm1.4}$ | $46.8_{\pm0.2}$ | $44.0_{\pm1.5}$ | $45.5_{\pm0.7}$ | |
| | 0.20% | $43.1_{\pm0.2}$ | $41.5_{\pm0.7}$ | $41.4_{\pm0.8}$ | $48.8_{\pm0.1}$ | $42.8_{\pm0.6}$ | $43.9_{\pm0.6}$ | $44.4_{\pm1.4}$ | $46.8_{\pm0.3}$ | $41.4_{\pm5.6}$ | $46.1_{\pm0.6}$ | |
| | 0.50% | $43.1_{\pm0.3}$ | $44.4_{\pm0.3}$ | $42.6_{\pm0.7}$ | $49.1_{\pm0.3}$ | $48.6_{\pm0.3}$ | $45.0_{\pm0.4}$ | $44.7_{\pm0.6}$ | $45.5_{\pm0.9}$ | $46.5_{\pm0.1}$ | $47.1_{\pm0.1}$ | |
| | 1.00% | $43.0_{\pm0.4}$ | $44.5_{\pm0.6}$ | $43.2_{\pm0.2}$ | $49.3_{\pm0.1}$ | $49.6_{\pm0.5}$ | $46.0_{\pm0.3}$ | $45.1_{\pm0.5}$ | $47.2_{\pm0.2}$ | $46.6_{\pm0.0}$ | $47.0_{\pm0.4}$ | |
| | 5.00% | $45.2_{\pm0.3}$ | $44.7_{\pm0.4}$ | $45.5_{\pm0.2}$ | OOM | OOM | OOM | OOM | OOM | $45.7_{\pm0.6}$ | $47.1_{\pm0.3}$ | |
| *ACM* | .003% | $83.9_{\pm1.7}$ | $82.5_{\pm1.6}$ | $76.3_{\pm2.4}$ | $84.8_{\pm0.3}$ | $84.8_{\pm0.3}$ | $89.3_{\pm0.6}$ | $80.6_{\pm0.8}$ | $90.3_{\pm2.3}$ | $92.2_{\pm0.2}$ | $73.4_{\pm0.5}$ | $91.7_{\pm0.4}$ |
| | .007% | $84.7_{\pm1.0}$ | $84.5_{\pm0.8}$ | $80.9_{\pm1.6}$ | $87.1_{\pm0.2}$ | $87.1_{\pm0.2}$ | $89.8_{\pm0.0}$ | $81.9_{\pm1.6}$ | $91.2_{\pm1.9}$ | $91.6_{\pm1.2}$ | $73.4_{\pm0.5}$ | |
| | .013% | $86.8_{\pm0.9}$ | $88.6_{\pm0.6}$ | $87.1_{\pm1.2}$ | $90.4_{\pm0.0}$ | $90.4_{\pm0.0}$ | $89.9_{\pm0.1}$ | $80.4_{\pm3.4}$ | $91.9_{\pm3.4}$ | $92.0_{\pm0.4}$ | $79.3_{\pm3.0}$ | |
| | .033% | $87.7_{\pm1.4}$ | $88.3_{\pm1.0}$ | $88.6_{\pm0.7}$ | $91.6_{\pm0.2}$ | $91.6_{\pm0.2}$ | $90.9_{\pm0.1}$ | $89.0_{\pm1.2}$ | $91.2_{\pm4.8}$ | $92.2_{\pm0.2}$ | $82.8_{\pm1.0}$ | |
| | .066% | $89.1_{\pm0.6}$ | $87.9_{\pm1.1}$ | $88.8_{\pm1.0}$ | $91.6_{\pm0.2}$ | $91.6_{\pm0.2}$ | $91.4_{\pm0.2}$ | $86.6_{\pm2.1}$ | $91.9_{\pm2.0}$ | $91.8_{\pm1.0}$ | $71.1_{\pm0.6}$ | |
| | .332% | $89.3_{\pm0.9}$ | $89.4_{\pm0.6}$ | $88.7_{\pm0.9}$ | $91.3_{\pm0.2}$ | $91.3_{\pm0.2}$ | $91.0_{\pm0.5}$ | $89.3_{\pm0.6}$ | $89.4_{\pm0.4}$ | $91.8_{\pm2.3}$ | $80.9_{\pm2.6}$ | |
| *DBLP* | .002% | $46.6_{\pm2.6}$ | $58.9_{\pm2.7}$ | $54.7_{\pm2.6}$ | $61.6_{\pm0.7}$ | $56.6_{\pm1.1}$ | $71.6_{\pm1.8}$ | $71.5_{\pm2.2}$ | $77.8_{\pm0.2}$ | $81.9_{\pm2.2}$ | $72.3_{\pm3.1}$ | $80.1_{\pm0.9}$ |
| | .004% | $57.0_{\pm1.4}$ | $62.9_{\pm0.9}$ | $54.9_{\pm2.9}$ | $60.4_{\pm1.1}$ | $62.0_{\pm1.5}$ | $75.4_{\pm2.6}$ | $76.2_{\pm2.1}$ | $80.9_{\pm1.4}$ | $81.7_{\pm3.5}$ | $72.3_{\pm0.5}$ | |
| | .007% | $67.5_{\pm0.8}$ | $61.3_{\pm1.6}$ | $60.9_{\pm0.4}$ | $70.7_{\pm0.3}$ | $68.3_{\pm0.6}$ | $71.7_{\pm4.4}$ | $73.6_{\pm0.7}$ | $81.1_{\pm0.6}$ | $81.2_{\pm0.4}$ | $67.3_{\pm1.0}$ | |
| | .019% | $68.0_{\pm1.1}$ | $65.7_{\pm1.2}$ | $64.5_{\pm1.1}$ | $73.6_{\pm0.2}$ | $73.5_{\pm0.4}$ | $76.6_{\pm0.6}$ | $74.6_{\pm1.1}$ | $77.9_{\pm0.3}$ | $80.8_{\pm2.1}$ | $72.0_{\pm0.9}$ | |
| | .037% | $67.6_{\pm1.5}$ | $65.0_{\pm1.1}$ | $66.6_{\pm1.5}$ | $74.6_{\pm0.1}$ | $75.2_{\pm0.1}$ | $76.9_{\pm1.8}$ | $76.4_{\pm0.8}$ | $78.5_{\pm0.6}$ | $81.9_{\pm3.2}$ | $71.6_{\pm2.3}$ | |
| | .186% | $65.7_{\pm1.0}$ | $66.3_{\pm0.8}$ | $66.0_{\pm1.0}$ | OOM | OOM | OOM | OOM | OOM | $82.1_{\pm0.3}$ | $69.9_{\pm0.9}$ | |

*(Left margin group labels: Homogeneous covers Cora, Reddit, Citeseer, ogbn-arxiv, Flickr; Heterogeneous covers ACM, DBLP.)*

# 3 Experimental Results and Analysis

## 3.1 Performance Comparison (RQ1)

We evaluate the GC methods across a spectrum of condensation ratios[2] to identify their progress and effective ranges. Prior studies primarily utilize three ratios of labeling rates, a selection that is far from comprehensive. We broaden the evaluation of existing condensation methods and the node-level and graph-level GC results are shown in Table 2 and Table 3, respectively. The experiment setting and additional results can be found in Appendix B.1.

For node-level experiments (Table 2), we observe that: (1) GC methods can achieve lossless results [45] compared to the whole dataset in 5 out of 7 cases (highlighted in red), generally outperforming traditional core-set methods, especially at smaller condensation ratios (e.g., *Citeseer* with $r$=0.18%, *ACM* with $r$=0.003%, *DBLP* with $r$=0.002%). (2) Distribution matching methods underperform compared to gradient matching and trajectory matching methods in 6 out of 7 datasets. The gradient matching methods and the trajectory matching methods perform well in our benchmark.

From graph-level experiments on GIN (Table 3) and GCN (Section B.1), our observations are: (1) *KiDD* with GIN shows significant advantages in 18 out of 25 cases, while *DosCond* and *Mirage* do not consistently outperform traditional core-set methods, indicating room for improvement in future work. (2) *KiDD* performs well when *GIN* is used as the model for downstream tasks but performs

---

[2]The condensation ratio is defined as $r = N'/N$ for node-level experiments and $r = n'/n$ for graph-level experiments. We use *Graph/Cls* to denote the number of condensed graphs per class in graph-level experiments.

Table 3: **Graph classification performance on GIN** (mean±std) across datasets with varying condensation ratios $r$. The best results are shown in **bold** and the runner-ups are shown in underlined. Red color highlights entries that exceed the whole dataset values.

| Dataset | Graph /Cls | Ratio($r$) | Traditional Core-set methods | | | Gradient | KRR | CTC | Whole Dataset |
|---|---|---|---|---|---|---|---|---|---|
| | | | **Random** | **Herding** | **K-Center** | **DosCond** | **KiDD** | **Mirage** | |
| *NCI1* Acc. (%) | 1 | 0.06% | $50.90_{\pm2.10}$ | $\underline{51.90}_{\pm1.60}$ | $\underline{51.90}_{\pm1.60}$ | $49.20_{\pm1.10}$ | $\mathbf{61.40}_{\pm0.50}$ | $50.80_{\pm2.20}$ | $80.0_{\pm1.8}$ |
| | 5 | 0.24% | $52.10_{\pm1.00}$ | $\underline{60.50}_{\pm2.40}$ | $47.00_{\pm1.10}$ | $51.10_{\pm0.80}$ | $\mathbf{63.20}_{\pm0.20}$ | $51.30_{\pm1.10}$ | |
| | 10 | 0.49% | $55.60_{\pm1.90}$ | $\underline{61.80}_{\pm1.50}$ | $49.40_{\pm1.80}$ | $50.30_{\pm1.30}$ | $\mathbf{64.20}_{\pm0.10}$ | $51.70_{\pm1.40}$ | |
| | 20 | 0.97% | $58.70_{\pm1.40}$ | $\underline{60.90}_{\pm1.90}$ | $55.20_{\pm1.60}$ | $50.30_{\pm1.30}$ | $\mathbf{60.90}_{\pm0.70}$ | $52.10_{\pm2.20}$ | |
| | 50 | 2.43% | $61.10_{\pm1.20}$ | $59.00_{\pm1.50}$ | $\underline{62.70}_{\pm1.50}$ | $50.30_{\pm1.30}$ | $\mathbf{65.40}_{\pm0.60}$ | $52.40_{\pm2.70}$ | |
| *DD* Acc. (%) | 1 | 0.21% | $49.70_{\pm11.30}$ | $58.80_{\pm6.10}$ | $58.80_{\pm6.10}$ | $46.30_{\pm8.50}$ | $\underline{71.30}_{\pm1.50}$ | $\mathbf{74.00}_{\pm0.40}$ | $70.1_{\pm2.2}$ |
| | 5 | 1.06% | $40.80_{\pm4.30}$ | $\underline{58.70}_{\pm5.80}$ | $51.30_{\pm5.30}$ | $57.50_{\pm5.60}$ | $70.90_{\pm1.10}$ | - | |
| | 10 | 2.12% | $63.10_{\pm5.20}$ | $\underline{64.10}_{\pm5.80}$ | $53.40_{\pm3.10}$ | $46.30_{\pm8.50}$ | $71.50_{\pm0.50}$ | - | |
| | 20 | 4.25% | $56.40_{\pm4.30}$ | $\underline{67.00}_{\pm2.60}$ | $58.50_{\pm5.70}$ | $40.70_{\pm0.00}$ | $71.20_{\pm0.90}$ | - | |
| | 50 | 10.62% | $58.90_{\pm6.30}$ | $\underline{68.40}_{\pm4.00}$ | $62.30_{\pm2.50}$ | $44.00_{\pm6.70}$ | $71.80_{\pm1.00}$ | - | |
| *ogbg-molbace* ROC-AUC | 1 | 0.17% | $0.468_{\pm.045}$ | $0.486_{\pm.035}$ | $0.486_{\pm.035}$ | $0.512_{\pm.092}$ | $\mathbf{0.706}_{\pm.000}$ | $\underline{0.590}_{\pm.004}$ | $0.763_{\pm.020}$ |
| | 5 | 0.83% | $0.312_{\pm.019}$ | $0.470_{\pm.042}$ | $0.553_{\pm.024}$ | $\underline{0.555}_{\pm.079}$ | $\mathbf{0.562}_{\pm.000}$ | $0.419_{\pm.010}$ | |
| | 10 | 1.65% | $0.442_{\pm.028}$ | $0.532_{\pm.031}$ | $\underline{0.594}_{\pm.019}$ | $0.536_{\pm.072}$ | $\mathbf{0.594}_{\pm.000}$ | $0.419_{\pm.010}$ | |
| | 20 | 3.31% | $0.510_{\pm.023}$ | $0.509_{\pm.052}$ | $\underline{0.512}_{\pm.031}$ | $0.484_{\pm.080}$ | $\mathbf{0.640}_{\pm.011}$ | $0.423_{\pm.011}$ | |
| | 50 | 8.26% | $0.486_{\pm.020}$ | $\underline{0.625}_{\pm.026}$ | $0.595_{\pm.026}$ | $0.503_{\pm.084}$ | $\mathbf{0.723}_{\pm.011}$ | - | |
| *ogbg-molbbbp* ROC-AUC | 1 | 0.12% | $0.510_{\pm.013}$ | $0.532_{\pm.015}$ | $0.532_{\pm.015}$ | $0.546_{\pm.026}$ | $\mathbf{0.616}_{\pm.000}$ | $\underline{0.592}_{\pm.004}$ | $0.635_{\pm.017}$ |
| | 5 | 0.61% | $0.522_{\pm.014}$ | $0.546_{\pm.020}$ | $\underline{0.581}_{\pm.022}$ | $0.519_{\pm.041}$ | $\mathbf{0.607}_{\pm.005}$ | $0.431_{\pm.013}$ | |
| | 10 | 1.23% | $0.508_{\pm.018}$ | $0.578_{\pm.017}$ | $\underline{0.619}_{\pm.027}$ | $0.505_{\pm.028}$ | $\mathbf{0.663}_{\pm.000}$ | $0.465_{\pm.036}$ | |
| | 20 | 2.45% | $0.567_{\pm.010}$ | $0.533_{\pm.009}$ | $0.546_{\pm.012}$ | $0.493_{\pm.031}$ | $\mathbf{0.677}_{\pm.001}$ | $\underline{0.610}_{\pm.022}$ | |
| | 50 | 6.13% | $\underline{0.595}_{\pm.014}$ | $0.552_{\pm.018}$ | $0.594_{\pm.016}$ | $0.509_{\pm.015}$ | $\mathbf{0.684}_{\pm.009}$ | $0.590_{\pm.031}$ | |
| *ogbg-molhiv* ROC-AUC | 1 | 0.01% | $0.366_{\pm.087}$ | $0.462_{\pm.072}$ | $0.462_{\pm.072}$ | $\underline{0.674}_{\pm.131}$ | $0.664_{\pm.016}$ | $\mathbf{0.710}_{\pm.009}$ | $0.701_{\pm.028}$ |
| | 5 | 0.03% | $0.501_{\pm.051}$ | $0.496_{\pm.044}$ | $0.519_{\pm.096}$ | $0.369_{\pm.175}$ | $\underline{0.657}_{\pm.005}$ | $\mathbf{0.703}_{\pm.012}$ | |
| | 10 | 0.06% | $\underline{0.554}_{\pm.031}$ | $0.458_{\pm.058}$ | $0.471_{\pm.054}$ | $0.457_{\pm.214}$ | $\mathbf{0.632}_{\pm.000}$ | $0.513_{\pm.055}$ | |
| | 20 | 0.12% | $0.621_{\pm.022}$ | $0.582_{\pm.027}$ | $0.627_{\pm.050}$ | $0.281_{\pm.007}$ | $\mathbf{0.648}_{\pm.025}$ | $\underline{0.633}_{\pm.048}$ | |
| | 50 | 0.30% | $\underline{0.625}_{\pm.062}$ | $0.600_{\pm.034}$ | $\mathbf{0.680}_{\pm.049}$ | $0.455_{\pm.214}$ | $0.587_{\pm.038}$ | $0.588_{\pm.067}$ | |

*Mirage cannot directly generate graphs with the required ratio. Parameter search aligns generated graphs with *DosCond* disk usage (see Appendix B.1). '-' denotes results unavailable due to recursive limits reached in MP Tree search.

poorly with *GCN*. This is because *KiDD* does not rely on the backbone and depends solely on the structure. Consequently, the stronger the downstream model's expressive ability, the better the results.

From both node-level and graph-level results, we observe that as the condensation ratio increases, traditional core-set methods improve, narrowing the performance gap with deep methods. However, deep GC methods show a saturation point or even a decline in performance beyond a certain threshold, suggesting that larger condensed data may introduce noise and biases that degrade performance.

**Key Takeaways 1:** Current node-level GC methods can achieve nearly lossless condensation performance. However, there is still a significant gap between graph-level GC and whole dataset training, indicating there is substantial room for improvement.

**Key Takeaways 2:** A large condensation ratio does not necessarily lead to better performance with current methods.

## 3.2 Structure in Graph Condensation (RQ2)

We analyze the impact of structure in terms of heterogeneity and heterophily. Experimental settings and additional results can be found in Appendix B.2.

(1) *Heterogeneity v.s. Homogeneity.* For the heterogeneous datasets *ACM* and *DBLP*, we convert the heterogeneous graphs into homogeneous ones for evaluation. From the results in Table 2, we observe that GC methods designed for homogeneous graphs preserve most of the semantic information and perform comparably to models training on the whole dataset.

(2) *Heterophily v.s. Homophily.* From the results of the heterophilous dataset *Flickr* (with homophily ratio $\mathcal{H} = 0.24$) in Table 2, we can observe that current GC methods can achieve almost the same accuracy as models training on the whole dataset. However, there is still a significant gap compared to the state-of-the-art results of the model designed for heterophilic graphs.

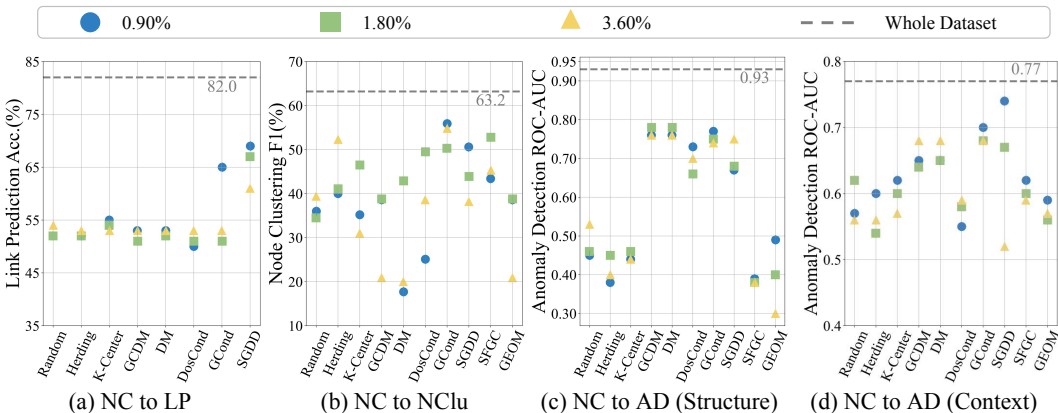

Figure 2: **Cross-task performance on *Citeseer***. For all downstream tasks, the models are trained solely using data of graphs condensed by node classification. For anomaly detection (c, d), structural and contextual anomalies [3] are injected into both the condensed graph and the original graph.

**Key Takeaways 3:** Existing GC methods primarily address simple graph data. However, the conversion process to specific data types is non-trivial, leaving significant room for improvement in preserving complex structural properties.

### 3.3 Transferability on Different Tasks (RQ3)

To evaluate the transferability of GC methods, we condense the dataset by node classification (NC) and use the condensed dataset to train models for link prediction (LP), node clustering (NClu), and anomaly detection (AD) tasks. The results on *Citeseer* are shown in Figure 2. Settings and additional results can be found in Appendix B.3.

As shown in Figure 2, performance with condensed datasets was significantly lower compared to original datasets in all transferred tasks. This decline may be due to the task-specific nature of the condensation process, which retains only task-relevant information while ignoring other semantically rich details. For instance, AD task prioritizes high-frequency graph signals more than NC and LP tasks, leading to poor performance when transferring condensed datasets from NC to AD tasks. Among the methods, gradient matching methods (*GCond*, *DosCond*, and *SGDD*) demonstrated better transferability in downstream tasks. In contrast, while structure-free methods (*SFGC* and *GEOM*) perform well in node classification (Section 3.1), they show a significant performance gap in AD tasks compared to gradient matching methods.

**Key Takeaways 4**: All condensed datasets struggle to perform well outside the context of the specific tasks for which they were condensed, leading to limited applicability.

### 3.4 Transferability of Backbone Model Architectures (RQ4)

We adopt one model (*SGC* or Graph Transformer) as the backbone for condensation and use the various models in downstream tasks evaluation. Details and additional results are in Appendix B.4.

As shown in Figure 3(a) and 3(b), each column shows the generalization performance of a condensed graph generated by different methods for various downstream models. We can observe that datasets condensed with *SGC* generally maintain performance when transferred across models. However, datasets condensed with *Graph Transformer* (*GTrans*) consistently underperform across various methods, and other models also exhibit reduced performance when adapted to *Graph Transformer*. Intuitively, *SGC*'s basic neighbor message-passing strategy may overlook global dependencies critical to more complex models, and similarly, complex models may not perform well when adapted to simpler models. As we can observe, *DosCond* exhibits generally better transferability compared to other gradient-matching methods. Since it can be regarded as the one-step gradient matching variant of *GCond*, we further test the impact of gradient matching steps on transferability (Figure 3(c)). Increasing the number of matching steps was found to correlate with reduced performance across architectures, indicating that extensive gradient matching may encode model-specific biases.

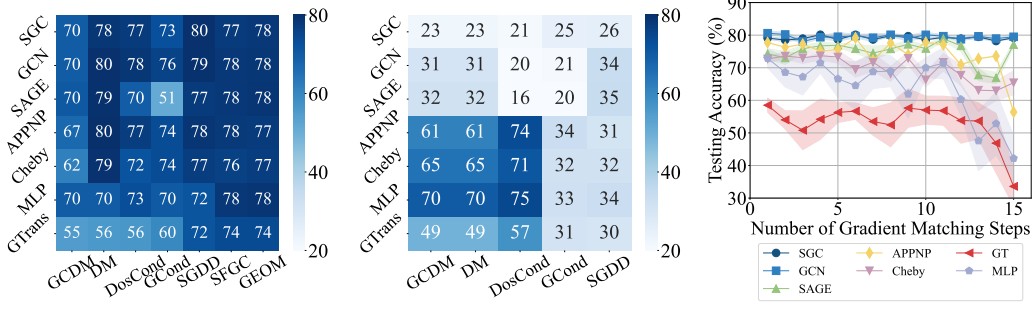

(a) Cross-arch. Acc. from *SGC*    (b) Cross-arch. Acc. from *GTrans*    (c) *GCond* across different steps.

Figure 3: **Cross-architecture performance.** Using *SGC* and Graph Transformer (*GTrans*) to condense *Cora* with a 2.6% ratio, we then test the accuracy on various downstream architectures (a, b). Furthermore, we evaluate the influence of gradient matching steps on *GCond* (c).

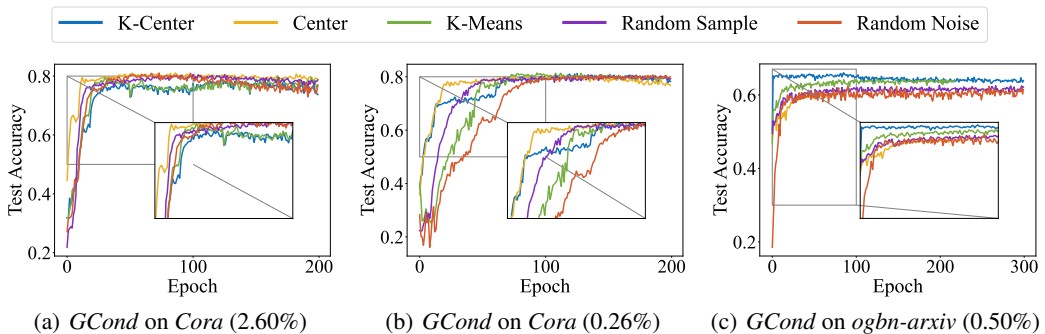

(a) *GCond* on *Cora* (2.60%)    (b) *GCond* on *Cora* (0.26%)    (c) *GCond* on *ogbn-arxiv* (0.50%)

Figure 4: **The impact of initialization** under different condensation ratios (a, b) and the impact across different datasets *Cora* (a, b) and *ogbn-arxiv* (c).

**Key Takeaways 5:** Current GC methods exhibit significant performance variability when transferred to different backbone architectures. Involving the entire training process potentially may lead to encoding backbone-specific details in the condensed datasets.

**Key Takeaways 6:** Despite their strong performance in general graph learning tasks, transformers surprisingly yield suboptimal results in graph condensation.

### 3.5 Initialization Impact (RQ5)

We evaluate 5 distinct initialization strategies, namely: *Random Noise*, *Random Sample*, *Center*, *K-Center*, and *K-Means*. The results of *GCond* on *Cora* and *ogbn-arxiv* are shown in Figure 4. Detailed settings and additional results can be found in Appendix B.5.

As shown in Figure 4(a) and Figure 4(b), the choice of the initialization method can significantly influence the efficiency of the condensation process but with little impact on the final accuracy. For instance, using *Center* on *Cora* reduces the average time to reach the same accuracy by approximately 25% compared to *Random Sample* and 71% compared to *Random Noise*. However, this speed advantage diminishes as the scale of the condensed graph increases. Additionally, different datasets have their preferred initialization methods for optimal performance. For example, *Center* is generally faster for *Cora* condensed by *GCond* while *K-Means* performs better on *ogbn-arxiv*.

**Key Takeaways 7:** Different datasets have their preferred initialization methods for optimal performance even for the same GC method.

**Key Takeaways 8:** The initialization mechanism primarily affects the convergence speed with little impact on the final performance. The smaller the condensed graph, the greater the influence of different initialization strategies on the convergence speed.

### 3.6 Efficiency and Scalability (RQ6)

In this subsection, we evaluate the condensation time and memory consumption of GC methods. The results on *ogbn-arxiv* are shown in Figure 5, where the *x*-axis denotes the overall condensation time (min) when achieving the best validation performance, the *y*-axis denotes the test accuracy (%), the inner size of the marker represents the peak CPU memory usage (MB), while the outer size represents the peak GPU memory usage (MB). As we can observe, the gradient matching methods have higher time and space consumption compared to other types of methods. However, Table 2 shows that current gradient and distribution matching GC methods may trigger OOM (Out of Memory) errors on large datasets with high con-

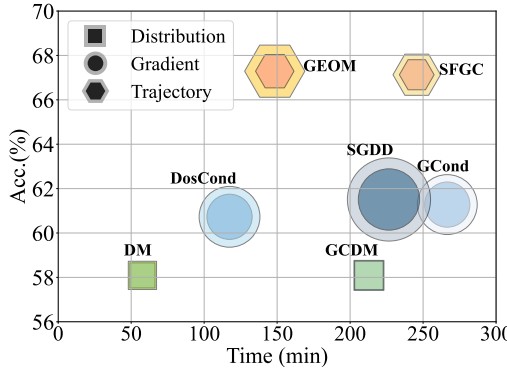

Figure 5: Time and memory consumption of different methods on *ogbn-arxiv* (0.50%).

densation ratios, making them unsuitable for large-scale scenarios, which contradicts the goal of applying graph condensation to extremely large graphs. More detailed results in Appendix B.6.

**Key Takeaways 9**: GC methods that rely on backbones and full-scale data training have large time and space consumption.

## 4 Future Directions

Notwithstanding the promising results, there are some directions worthy to explore in the future:

**Theory of optimal condensation.** According to our findings, GC methods are striving to achieve better performance with smaller condensed dataset sizes but it's not necessarily true that larger compressed datasets lead to better results. How to trade off between dataset size, information condensation, and information preservation, and whether there exists a theory of Pareto-optimal condensation in the graph condensation process, are future research directions.

**Condensation for more complex graph data.** Current GC methods are predominantly tailored to the simplest types of graphs, overlooking the diversity of graph structures such as heterogeneous graphs, directed graphs, hypergraphs, signed graphs, dynamic graphs, text-rich graphs, etc. There is a pressing need for research on graph condensation methods that cater to more complex graph data.

**Task-Agnostic graph condensation.** Task-agnostic GC methods could greatly enhance flexibility and utilization in graph data analysis, promoting versatility across various domains. Current methods often depend on downstream labels or task-specific training. Future research should focus on developing task-agnostic, unsupervised, or self-supervised GC methods that preserve crucial structural and semantic information independently of specific tasks or datasets.

**Improving the efficiency and scalability of graph condensation methods.** Efficient and scalable GC methods are crucial yet challenging to design. Most current methods combine condensation with full training, making them resource-heavy and less scalable. Decoupling these processes could significantly enhance GC's efficiency and scalability, broadening its use across various domains.

## 5 Conclusion and Future Works

This paper introduces a comprehensive graph condensation benchmark, GC-Bench, by integrating and comparing 12 methods across 12 datasets covering varying types and scopes. We conduct extensive experiments to reveal the performance of GC methods in terms of effectiveness, transferability, and efficiency. We implement an library (https://github.com/RingBDStack/GC-Bench) that incorporates all the aforementioned protocols, baseline methods, datasets, and scripts to reproduce the results in this paper. The GC-Bench library offers a comprehensive and unbiased platform for evaluating current methods and facilitating future research. In this study, we mainly evaluate the performance of GC methods for the node classification and graph classification task, which is widely adopted in the previous literature. In the future, we plan to extend the GC-Bench with broader coverage of datasets and tasks, providing further exploration of the generalization ability of GC methods. We will update the benchmark regularly to reflect the most recent progress in GC methods.

## Acknowledgements

The corresponding author is Jianxin Li. This work is supported by the NSFC through grants No.62225202 and No.62302023, the Fundamental Research Funds for the Central Universities, CAAI-MindSpore Open Fund, developed on OpenI Community. This work is also supported in part by NSF under grants III-2106758, and POSE-2346158.

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

# A Details of GC-Bench

## A.1 Datasets

The evaluation node-level datasets include 5 homogeneous datasets (3 transductive datasets, i.e., *Cora*, *Citeseer* [16] and *ogbn-arxiv* [13], and 2 inductive datasets, i.e., *Flickr* [43] and *Reddit* [12]) and 2 heterogeneous datasets (*ACM* [46] and *DBLP* [7]). The evaluation graph-level datasets include 5 datasets ( *NCI1* [32], *DD* [4], *ogbg-molbace* [13], *ogbg-molhiv* [13], *ogbg-molbbbp* [13]).

We utilize the standard data splits provided by PyTorch Geometric [6] and the Open Graph Benchmark (OGB) [13] for our experiments. For datasets in TUDataset [27], we split the data into 10% for testing, 10% for validation, and 80% for training. For *ACM* and *DBLP* datasets, we follow the settings outlined in [25]. Dataset statistics are shown in Table A1.

Table A1: Dataset statistics. For heterogeneous datasets, the features are from the target nodes (papers in *ACM* and authors in *DBLP*).

|  | Dataset | #Nodes / #Avg. Nodes | #Edges / #Avg. Edges | #Classes | #Features / Graphs |
|---|---|---|---|---|---|
| **Node-level** | *Cora* | 2,708 | 5,429 | 7 | 1,433 |
| | *Citeseer* | 3,327 | 4,732 | 6 | 3,703 |
| | *ogbn-arxiv* | 169,343 | 1,166,243 | 40 | 128 |
| | *Flickr* | 89,250 | 899,756 | 7 | 500 |
| | *Reddit* | 232,965 | 57,307,946 | 210 | 602 |
| | *ACM* | 10,942 | 547,872 | 3 | 1,902 |
| | *DBLP* | 37,791 | 170,794 | 4 | 334 |
| **Graph-level** | *ogbg-molhiv* | 25.5 | 54.9 | 2 | 41,127 |
| | *ogbg-molbace* | 34.1 | 36.9 | 2 | 1,513 |
| | *ogbg-molbbbp* | 24.1 | 26.0 | 2 | 2,039 |
| | *NCI1* | 29.8 | 32.3 | 2 | 4,110 |
| | *DD* | 284.3 | 715.7 | 2 | 1,178 |

## A.2 Algorithms

We summarize the current GC algorithms in Table A2. We choose **12** representative ones for evaluation in this paper including *Random*, *K-Center* [30], *Herding* [36], *GCond* [15], *DosCond* [14], *SGDD* [42], *GCDM* [20], *DM* [24], *SFGC* [47], *GEOM* [45], *KiDD* [41], *Mirage* [11]. **We will continue to update and improve the benchmark to include more algorithms.** Here we introduce the GC algorithms in detail:

- **Traditional Core-set Methods**
  - **Random**: For node classification tasks, nodes are randomly selected to form a new subgraph. For graph classification, the graphs are randomly selected to create a new subset.
  - **Herding** [36]: The nodes or graphs are selected samples that are closest to the cluster center.
  - **K-Center** [30]: Nodes or graphs are chosen such that they have the minimal distance to the nearest cluster center, which is generated using the K-Means Clustering method.

- **Gradient Matching Methods**
  - **GCond** [15]: In GCond, the optimization of the synthetic dataset is framed as a bi-level problem. It adapts a gradient matching scheme to match the gradients of GNN parameters between the condensed and original graphs, while optimizing the model's performance on the datasets. For generating the synthetic adjacency matrix, GCond employs a Multi-Layer Perceptron (MLP) to model the edges by using node features as input, maintaining the correlations between node features and graph structures.
  - **DosCond** [14]: In DosCond, the gradient matching scheme only matches the network gradients for model initialization $\theta_0$ while discarding the training trajectory of $\theta_t$, which accelerated the entire condensation process by only informing the direction to update the

Table A2: Summary of Graph Condensation (GC) algorithms. We also provide public access to the official algorithm implementations. "KRR" is short for Kernel Ridge Regression and "CTC" is short for computation tree compression. "GNN" is short for Graph, "GNTK" is short for graph neural tangent kernel, "SD" is short for spectral decomposition. "NC" is short for node classification, "LP" is short for link prediction, "AD" is short for anomaly detection, and "GC" is short for graph classification.

| Taxonomy | Method | Initialization | Backbone Model | Downstream Task | Code | Venue |
|---|---|---|---|---|---|---|
| Traditional Methods | Random | — | — | — | — | — |
| | Herding [36] | — | — | — | link | ICML, 2009 |
| | K-Center [30] | — | — | — | link | ICLR, 2018 |
| Gradient Matching | GCond [15] | Random Sample | GNN | NC | link | ICLR, 2021 |
| | DosCond [14] | Random Sample | GNN | NC, GC | link | SIGKDD, 2022 |
| | MSGC [8] | Zero Matrix | GNN | NC | — | KBS, 2023 |
| | SGDD [42] | Random Sample | GNN | NC, LP, AD | link | NeurIPS, 2023 |
| | GCARe [26] | — | GNN | NC | — | Appl. Sci. 2023 |
| | CTRL [44] | K-Means | GNN | NC, GC | link | arXiv, 2024 |
| | GroC [19] | Random Sample | GNN | NC, GC | — | arXiv, 2023 |
| | EXGC [5] | Random Sample | GNN | NC | link[1] | WWW 2024 |
| | MCond [9] | Random Sample | GNN | NC | — | ICDE, 2024 |
| Distribution Matching | GCDM [20] | Random Sample | GNN | NC | — | arXiv, 2022 |
| | DM [22, 24] | Random Sample | GNN | NC | — | ICDM, 2023 |
| | GDEM [21] | Eigenbasis Approximation | SD | NC | link | ICML, 2024 |
| | FedGKD [28] | Random Noise | GNN | NC | — | arXiv, 2023 |
| Trajectory Matching | SFGC [47] | K-Center | GNN | NC | link | NeurIPS, 2023 |
| | GEOM [45] | K-Center | GNN | NC | link | ICML, 2024 |
| KRR | GC-SNTK [34] | Random Noise | GNTK | NC | link | WWW, 2024 |
| | KiDD [41] | Random Sample | GNTK | GC | link | SIGKDD, 2023 |
| CTC | Mirage [11] | — | GNN | GC | link | ICLR, 2024 |

[1] The code repository for EXGC is not fully developed.

synthetic dataset. DosCond also modeled the discrete graph structure as a probabilistic model and each element in the adjacency matrix follows a Bernoulli distribution.

− **MSGC** [8]: MSGC condenses a large-scale graph into multiple small-scale sparse graphs, leveraging neighborhood patterns as substructures to enable the construction of various connection schemes. This process enriches the diversity of embeddings generated by GNNs, enhances the representation power of GNNs con complex graphs.

− **SGDD** [42]: SGDD uses graphon approximation to ensure that the structural information of the original graph is retained in the synthetic, condensed graph. The condensed graph structure is optimized by minimizing the optimal transport (OT) distance between the original structure and the condensed structure.

− **GCARe** [26]: GCARe addresses biases in condensed graphs by regularizing the condensation process, ensuring that the knowledge of different subgroups is distilled fairly into the resulting graphs.

− **CTRL** [44]: CTRL clusters each class of the original graph into sub-clusters and uses these as initial value for the synthetic graph. By considering both the direction and magnitude of gradients during gradient matching, it effectively minimizes matching errors during the condensation phase.

− **GroC** [19]: GroC uses an adversarial training (bi-level optimization) framework to explore the most impactful parameter spaces and employs a Shock Absorber operator to apply targeted adversarial perturbation.

− **EXGC** [5]: EXGC leverages Mean-Field variational approximation to address inefficiency in the current gradient matching schemes and uses the Gradient Information Bottleneck objective to tackle node redundancy.

− **MCond** [9]: MCond addresses the limitations of traditional condensed graphs in handling unseen data by learning a one-to-many node mapping from original nodes to synthetic nodes and uses an alternating optimization scheme to enhance the learning of synthetic graph and mapping matrix.

• **Distribution Matching Methods**

- **GCDM** [20]: GCDM synthesizes small graphs with receptive fields that share a similar distribution to the original graph, achieved through a distribution matching loss quantified by maximum mean discrepancy (MMD).
- **DM** [22, 24]: DM can be regarded as a one-step variant of GCDM. In DM, the optimization is centered on the initial parameters. Notably, in [22] and [24], DM does not learn any structures for efficiency. However, for better comparison in our experiments, we continue to learn the adjacency matrix.
- **FedGKD** [28]: FedGKD trains models on condensed local graphs within each client to mitigate the potential leakage of the training set membership. FedGKD features a Federated Graph Neural Network framework that enhances client collaboration using a task feature extractor for graph data distillation and a task relator for globally-aware model aggregation.

- **Trajectory Matching Methods**
  - **SFGC** [47]: SFGC uses trajectory matching instead of a gradient matching scheme. It first trains a set of GNNs on original graphs to acquire and store an expert parameter distribution offline. The expert trajectory guides the optimization of the condensed graph-free data. The generated graphs are evaluated using closed-form solutions of GNNs under the graph neural tangent kernel (GNTK) ridge regression, avoiding iterative GNN training.
  - **GEOM** [45]: GEOM makes the first attempt toward lossless graph condensation using curriculum-based trajectory matching. A homophily-based difficulty score is assigned to each node and the easy nodes are learned in the early stages while more difficult ones are learned in the later stages. On top of that, GEOM incorporated a Knowledge Embedding Extraction (KEE) loss into a matching loss.

- **Kernel Ridge Regression Methods**
  - **GC-SNTK** [34]: GC-SNTK introduces a Structure-based Neural Tangent Kernel(SNTK) to capture graph topology, replacing the inner GNNs training in traditional GC paradigm, avoiding multiple iterations.
  - **KiDD** [41]: KiDD uses kernel ridge regression (KRR) with a graph neural tangent kernel (GNTK) for graph-level tasks. To enhance efficiency, KiDD introduces LiteGNTK, a simplified GNTK, and proposes KiDD-LR for faster low-rank approximation and KiDD-D for handling discrete graph topology using the Gumbel-Max reparameterization trick. We use KiDD-LR for experiments as it has generally demonstrated better performance compared to KiDD-D.

- **Computation Tree Compression Methods**
  - **Mirage** [11]: Mirage decomposes graphs in datasets into a collection of computation trees and then mines frequently co-occurring trees from this set. Mirage then uses aggregation functions (MEANPOOL, SUMPOOL, etc.) on the embeddings of the root node of each tree to approximate the graph embedding.

## A.3   Hyper-Parameter Setting

For the implementation of various graph condensation methods, we adhere to the default parameters as specified by the authors in their respective original implementations. This approach ensures that our results are comparable to those reported in the foundational studies. For condensation ratios that were not explored in the original publications, we employ a grid search strategy to identify the optimal hyperparameters within the predefined search space. This includes experimenting with various combinations, such as differing learning rates for the feature optimizer and the adjacency matrix optimizer. The corresponding hyperparameter space are shown in Table A3.

## A.4   Computation resources

All experiments were conducted on a high-performance GPU cluster to ensure a fair comparison. The cluster consists of 32 identical dell-GPU nodes, each featuring 256GB of memory, 2 Intel Xeon processors, and 4 NVIDIA Tesla V100 GPUs, with each GPU having 64 GB of GPU memory. If any experiment setting exceeds the GPU memory limit, it is reported as out-of-memory (OOM).

Table A3: Hyperparameter search space of different methods

| Method | Hyperparameter | Values |
|---|---|---|
| General Settings | Learning Rate | 0.1, 0.01, 0.001, 0.0001, 0.00001 |
| | Epochs | 300, 400, 500, 800, 1000, 2000, 3000, 4000, 5000 |
| | Layers | 2, 3 |
| | Dropout Rate | 0, 0.05, 0.1, 0.5, 0.6, 0.7, 0.8 |
| | Weight Decay | 0, 0.0005 |
| | Hidden Units | 128, 256 |
| | Pooling | sum, mean |
| | Activation | LeakyReLU, ReLU, Sigmoid, Softmax |
| | Batch Size | (16,6000) |
| SGDD | mx_size | 50, 100 |
| | opt_scale | 5, 10 |
| GCond, DosCond, SGDD, GCDM, DM | outer loop | 1, 2, 5, 10, 15, 20 |
| GCond, SGDD, GCDM | inner loop | 1, 5, 10, 15, 20 |
| SFGC, GEOM | expert_epochs | 50, 70, 100, 350, 600, 800, 1000, 1500, 1600, 1900 |
| | start_epoch | 10, 20, 50, 100, 200, 300 |
| | teacher_epochs | 800, 1000, 1200, 2400, 3000 |
| GEOM | lam | 0.6, 0.7, 0.75, 0.8, 0.85, 0.9, 0.95 |
| | T | 250, 500, 600, 800, 1000, 1200 |
| | scheduler | linear, geom, root |
| KiDD | scale | uniform, degree |
| | rank | 8, 16, 32 |
| | orth_reg | 0.01, 0.001, 0.0001 |

## A.5 Discussion on Existing Benchmarks

To the best of our knowledge, the only concurrent work is GCondenser [23]. The comparison of GCondser and our GC-Bench are list in Table A4. GCondenser [23] focus the node-level GC methods for node classification on homogeneous graphs with limited evaluation dimensions in terms of performance and time efficiency. Our GC-Bench analyzes more GC methods on a wider variety of datasets (both homogeneous and heterogeneous) and tasks (node classification, graph classification), encompassing both node-level and graph-level methods. In addition to performance and efficiency analysis, we further explore the transferability across different tasks (link prediction, node clustering, anomaly detection) and backbones (GNN models and the popular Graph Transformer). With GC-Bench covering more in-depth investigation over a wider scope, we believe it will provide valuable insights into existing works and future directions.

# B  Settings and Additional Results

In this section, we provide more details of the experimental settings and the additional results for the proposed 6 research questions, respectively.

## B.1  Settings and Additional Results of Performance Comparison (RQ1)

### B.1.1  Comparison Setting

**Node Classification Graph Dataset Setting.**    We compared ten state-of-the-art GC methods. The selection of the condensation ratio $r$ is based on the labeling rates of different datasets. For datasets like *Cora* and *Citeseer*, the labeling rates are less than 50%, we select $r$ as a proportion of the labeling rate, specifically at $\{5\%, 10\%, 25\%, 50\%, 75\%, 100\%\}$. For datasets like *ogbn-arxiv*, and inductive datasets where all nodes in the training graphs are labeled, with a relatively higher labeling rate, $r$ is chosen to be $\{5\%, 10\%, 25\%, 50\%, 75\%, 100\%\}$. Corresponding condensation rates are shown in Table B2.

**Graph Classification Graph Dataset Setting.**    We compared three state-of-the-art GC algorithms on graph classification datasets: *DosCond* [14], *KiDD* [41], and *Mirage* [11]. *Mirage* [11] does not condense datasets into unified graphs measurable by Ǵraphs per Class(GPC) as *DosCond* [14] and

Table A4: Comparison of GCondenser and GC-Bench

| Benchmark Coverage | | GCondenser | GC-Bench |
|---|---|---|---|
| **Algorithms** | Traditional Core-set Methods | Random, K-Center | Random, K-Center, Herding |
| | Gradient Matching | GCond, DosCond, SGDD | GCond, DosCond, SGDD |
| | Distribution Matching | GCDM, DM | GCDM, DM |
| | Trajectory Matching | SFGC | SFGC, GEOM |
| | KRR | — | KiDD |
| | CTC | — | Mirage |
| **Datasets** | Node-level Homogenerous | *Cora*, *Citeseer*, *ogbn-arxiv* *Flickr*, *Reddit*, *PubMed* | *Cora*, *Citeseer*, *ogbn-arxiv* *Flickr*, *Reddit* |
| | Node-level Heterogenerous | — | *ACM*, *DBLP* |
| | Graph-level | — | *NCI1*, *DD*, *ogbg-molbace* *ogbg-molbbbp*, *ogbg-molhiv* |
| **Tasks** | Nodel-level | node classification | node classification link prediction node clustering anomaly detection |
| | Graph-level | — | graph classification |
| **Evaluation Dimensions** | Perf. — Condensation Ratios | ✓ | ✓ |
| | Perf. — Impact of Struture | structure v.s. structure-free | structure v.s. structure-free structure properties (Heterogeneity, Heterophily) |
| | Perf. — Impact of Initialization | ✓ | ✓ |
| | Trans. — Backbone Trans. | SGC and GCN transfer to SGC, GCN, GraphSAGE, APPNP, CHebyNet, MLP | SGC, GCN and Graph Transformer transfer to SGC, GCN, GraphSAGE, APPNP, ChebyNet, MLP, Graph Transformer |
| | Trans. — Task Trans. | — | node classification link prediction node clustering anomaly detection |
| | Efficiency — Time | ✓ | ✓ |
| | Efficiency — Space | — | ✓ |

*KiDD* [41] do. Therefore, we measure the condensed dataset size by storing its elements in .pt format, similar to *DosCond* [14] and *KiDD* [41]. We select the Mirage-condensed dataset size closest to *DosCond*'s as the corresponding GPC. *KiDD* [41] generally occupies more disk space than DosCond under the same GPC. The size of *Mirage* datasets is determined by two parameters: the number of GNN layers ($L$) and the frequency threshold $\Theta$. We fix $L = 2$, consistent with the 2-layer model used for validation, and employ a grid search strategy to identify the threshold combination that yields a dataset size closest to the targeted GPC. The corresponding disk space, GPC, and threshold choices are presented in Table B1. Note that for small thresholds, the MP Tree search algorithms used in *Mirage* [11] may reach recursive limits. Consequently, in *DD* and *ogbg-molbace*, certain GPCs lack corresponding threshold values.

**Heterogeneous Graph Dataset Setting.**   Due to the absence of condensation methods specifically for heterogeneous graphs, we convert heterogeneous datasets into homogeneous graphs for condensation, focusing on target nodes. We uniformly summed the adjacency matrices corresponding to various meta-paths as in [25], and applied zero-padding to match the maximum feature dimension as well as one-hot encoding for nodes without features. Specifically, in *GEOM* [45], when calculating heterophily, all nodes without labels (non-target nodes) are assigned the same distinct label, ensuring a consistent heterophily calculation.

### B.1.2   Additional Results

The graph classification performance on GCN is shown in Table B3. *DosCond* [14] with GCN demonstrates significant advantages in 12 out of 25 cases, while *KiDD* [41] underperforms in most scenarios. Notably, *DosCond* [14] and *Mirage* [11] even outperform the results of the whole dataset

Table B1: Comparison of Disk Size and Graph per Class (GPC) for condensed datasets between *Mirage* and *DosCond*.

| Dataset | Graph/ Cls | Mirage Disk Size (Bytes) | DosCond Disk Size (Bytes) | Class 0 Threshold | Class 1 Threshold |
|---|---|---|---|---|---|
| *NCI1* [32] | 1 | 14,455 | 18,425 | 451 | 441 |
| | 5 | 81,622 | 82,745 | 351 | 381 |
| | 10 | 142,228 | 162,301 | 301 | 291 |
| | 20 | 195,609 | 324,035 | 251 | 231 |
| | 50 | 995,277 | 806,403 | 201 | 171 |
| *DD* [4] | 1 | 38,352 | 855,077 | 15 | 9 |
| | 5 | — | 4,265,957 | — | — |
| | 10 | — | 8,529,583 | — | — |
| | 20 | — | 17,056,751 | — | — |
| | 50 | — | 42,638,383 | — | — |
| *ogbg-molbace* [13] | 1 | 13,836 | 14143 | 120 | 90 |
| | 5 | 60,047 | 60,927 | 230 | 80 |
| | 10 | 106,077 | 119,497 | 120 | 80 |
| | 20 | 232,191 | 236,489 | 140 | 70 |
| | 50 | — | 587,337 | — | — |
| *ogbg-molbbbp* [13] | 1 | 8,817 | 8,831 | 29 | 198 |
| | 5 | 34,699 | 34,175 | 49 | 109 |
| | 10 | 66,433 | 65,929 | 30 | 90 |
| | 20 | 104,091 | 129,289 | 20 | 80 |
| | 50 | 324,425 | 319,369 | 17 | 87 |
| *ogbg-molhiv* [13] | 1 | 9,606 | 9,717 | 8,000 | 250 |
| | 5 | 54,669 | 38,837 | 1,760 | 170 |
| | 10 | 74,524 | 75,263 | 1,680 | 130 |
| | 20 | 148,028 | 148,095 | 1,420 | 110 |
| | 50 | 330,498 | 366,463 | 800 | 110 |

Table B2: Different condensation ratios of transductive datasets. For heterogeneous datasets, the number of nodes in the original graph is the sum of all types of nodes.

| Ratio ($r$) | *Cora* | *Citeseer* | *ACM* | *DBLP* |
|---|---|---|---|---|
| 5% | 0.26% | 0.18% | 0.003% | 0.002% |
| 10% | 0.52% | 0.36% | 0.007% | 0.004% |
| 25% | 1.30% | 0.90% | 0.013% | 0.007% |
| 50% | 2.60% | 1.80% | 0.033% | 0.019% |
| 75% | 3.90% | 2.70% | 0.066% | 0.037% |
| 100% | 5.20% | 3.60% | 0.332% | 0.186% |

on *ogbg-molbace*. For *Mirage* [11], due to the algorithm's recursive depth under low threshold parameters, we have only one result corresponding to GPC 1 on *DD*. However, this single result already surpasses all datasets condensed by *KiDD* [41] and the dataset with GPC 1 condensed by *DosCond*.

## B.2 Settings and Additional Results of Structure in Graph Condensation (RQ2)

### B.2.1 Experimental Settings

The homophily ratio we use is the edge homophily ratio, which represents the fraction of edges that connect nodes with the same labels. It can be calculated as:

$$\mathcal{H}(G) = \frac{1}{|\mathcal{E}|} \sum_{(j,k)\in\mathcal{E}} \mathbf{1}(y_j = y_k), \ \ i \in \mathcal{V}, \tag{A.1}$$

where $\mathcal{V}$ is the node set, $\mathcal{E}$ is the edge set, $|\mathcal{E}|$ is the number of edges in the graph, $y_i$ is the label of node $i$ and $\mathbf{1}(\cdot)$ is the indicator function. A graph is typically considered to be highly homophilous when $\mathcal{H}$ is large (typically, $0.5 \leq \mathcal{H} \leq 1$), such as *Cora* and *Reddit*. Conversely, a graph with a low edge homophily ratio is considered to be heterophilous, such as *Flickr*.

Table B3: **Graph classification performance on GCN** (mean±std) across datasets with varying condensation ratios $r$. The best results are shown in **bold** and the runner-ups are shown in underlined. Red color highlights entries that exceed the whole dataset values.

| Dataset | Graph /Cls | Ratio($r$) | Traditional Core-set methods | | | Gradient | KRR | CTC | Whole Dataset |
|---|---|---|---|---|---|---|---|---|---|
| | | | Random | Herding | K-Center | DosCond | KiDD | Mirage | |
| NCI1 Acc. (%) | 1 | 0.06% | $53.30_{\pm0.6}$ | $55.20_{\pm2.6}$ | $55.20_{\pm2.6}$ | **$57.30_{\pm0.9}$** | $49.30_{\pm1.1}$ | $49.10_{\pm0.9}$ | $71.1_{\pm0.8}$ |
| | 5 | 0.24% | $55.00_{\pm1.4}$ | $56.50_{\pm0.9}$ | $53.20_{\pm0.6}$ | **$58.40_{\pm1.4}$** | $56.10_{\pm1.0}$ | $49.60_{\pm2.2}$ | |
| | 10 | 0.49% | $58.10_{\pm2.2}$ | **$58.60_{\pm0.8}$** | $57.00_{\pm2.6}$ | $57.80_{\pm1.6}$ | $57.50_{\pm1.1}$ | $48.60_{\pm0.1}$ | |
| | 20 | 0.97% | $54.40_{\pm0.8}$ | $59.10_{\pm1.1}$ | **$60.10_{\pm1.3}$** | $60.10_{\pm3.2}$ | $56.40_{\pm0.6}$ | $48.70_{\pm0.0}$ | |
| | 50 | 2.43% | $56.80_{\pm1.1}$ | $58.70_{\pm1.1}$ | **$64.40_{\pm0.9}$** | $58.20_{\pm2.8}$ | $59.90_{\pm0.6}$ | $48.60_{\pm0.1}$ | |
| DD Acc. (%) | 1 | 0.21% | $59.70_{\pm1.5}$ | $66.90_{\pm2.8}$ | $66.90_{\pm2.8}$ | $68.30_{\pm6.6}$ | $58.60_{\pm2.4}$ | **$71.20_{\pm6.6}$** | $78.4_{\pm1.7}$ |
| | 5 | 1.06% | $61.90_{\pm1.1}$ | $66.20_{\pm2.5}$ | $62.00_{\pm1.7}$ | **$73.10_{\pm2.2}$** | $58.60_{\pm1.1}$ | - | |
| | 10 | 2.12% | $63.70_{\pm2.8}$ | $68.00_{\pm3.6}$ | $62.50_{\pm2.3}$ | **$71.30_{\pm8.3}$** | $61.60_{\pm3.8}$ | - | |
| | 20 | 4.25% | $64.70_{\pm5.3}$ | $69.70_{\pm0.8}$ | $63.10_{\pm1.9}$ | **$73.00_{\pm5.8}$** | $62.60_{\pm1.4}$ | - | |
| | 50 | 10.62% | $66.60_{\pm2.1}$ | $68.50_{\pm1.4}$ | $68.90_{\pm1.8}$ | **$74.20_{\pm3.6}$** | $59.30_{\pm0.0}$ | - | |
| ogbg-molbace ROC-AUC | 1 | 0.17% | $0.510_{\pm.083}$ | $0.515_{\pm.040}$ | $0.517_{\pm.044}$ | $0.658_{\pm.064}$ | $0.568_{\pm.047}$ | **$0.733_{\pm.012}$** | $0.711_{\pm.019}$ |
| | 5 | 0.83% | $0.612_{\pm.036}$ | $0.653_{\pm.043}$ | $0.508_{\pm.087}$ | $0.691_{\pm.06}$ | $0.356_{\pm.022}$ | **$0.760_{\pm.002}$** | |
| | 10 | 1.65% | $0.620_{\pm.054}$ | $0.658_{\pm.046}$ | $0.646_{\pm.047}$ | $0.702_{\pm.045}$ | $0.542_{\pm.027}$ | **$0.759_{\pm.002}$** | |
| | 20 | 3.31% | $0.642_{\pm.053}$ | $0.631_{\pm.051}$ | $0.575_{\pm.03}$ | $0.659_{\pm.049}$ | $0.526_{\pm.014}$ | **$0.761_{\pm.003}$** | |
| | 50 | 8.26% | $0.677_{\pm.015}$ | $0.629_{\pm.053}$ | $0.576_{\pm.087}$ | **$0.714_{\pm.032}$** | $0.446_{\pm.042}$ | - | |
| ogbg-molbbbp ROC-AUC | 1 | 0.12% | $0.534_{\pm.041}$ | $0.560_{\pm.017}$ | $0.560_{\pm.017}$ | **$0.600_{\pm.023}$** | $0.504_{\pm.042}$ | **$0.600_{\pm.002}$** | $0.646_{\pm.013}$ |
| | 5 | 0.61% | $0.561_{\pm.014}$ | $0.574_{\pm.022}$ | $0.585_{\pm.005}$ | $0.579_{\pm.056}$ | $0.561_{\pm.004}$ | **$0.609_{\pm.061}$** | |
| | 10 | 1.23% | $0.566_{\pm.011}$ | $0.590_{\pm.024}$ | **$0.598_{\pm.025}$** | $0.556_{\pm.063}$ | $0.550_{\pm.005}$ | $0.517_{\pm.028}$ | |
| | 20 | 2.45% | $0.593_{\pm.023}$ | $0.568_{\pm.019}$ | $0.545_{\pm.009}$ | $0.590_{\pm.057}$ | $0.594_{\pm.022}$ | **$0.626_{\pm.032}$** | |
| | 50 | 6.13% | $0.587_{\pm.007}$ | $0.579_{\pm.022}$ | **$0.621_{\pm.011}$** | $0.598_{\pm.024}$ | $0.603_{\pm.01}$ | $0.602_{\pm.018}$ | |
| ogbg-molhiv ROC-AUC | 1 | 0.01% | $0.733_{\pm.008}$ | $0.727_{\pm.012}$ | $0.727_{\pm.012}$ | **$0.734_{\pm.002}$** | $0.725_{\pm.007}$ | $0.728_{\pm.012}$ | $0.750_{\pm.010}$ |
| | 5 | 0.03% | $0.729_{\pm.006}$ | $0.720_{\pm.018}$ | **$0.739_{\pm.01}$** | $0.736_{\pm.008}$ | $0.738_{\pm.003}$ | $0.717_{\pm.003}$ | |
| | 10 | 0.06% | $0.724_{\pm.011}$ | $0.726_{\pm.014}$ | $0.723_{\pm.012}$ | **$0.736_{\pm.007}$** | $0.731_{\pm.008}$ | $0.735_{\pm.028}$ | |
| | 20 | 0.12% | $0.723_{\pm.015}$ | $0.726_{\pm.015}$ | $0.724_{\pm.01}$ | **$0.733_{\pm.007}$** | $0.703_{\pm.097}$ | $0.710_{\pm.016}$ | |
| | 50 | 0.30% | $0.712_{\pm.014}$ | $0.723_{\pm.019}$ | $0.721_{\pm.012}$ | **$0.731_{\pm.011}$** | $0.723_{\pm.011}$ | $0.718_{\pm.022}$ | |

∗*Mirage* cannot directly generate graphs with the required ratio. Thus, we search the parameter space and aligned the generated graph to match *DosCond*'s disk usage as substitution (see Appendix B.1).

We also calculate the homophily ratio of condensed datasets. Since the condensed datasets have weighted edges, we first sparsify the graph by removing all edges with weights less than 0.05, then calculate the homophily ratio by adjusting the fraction to a weighted fraction, which can be represented as:

$$\mathcal{H}(G) = \frac{\sum_{(j,k)\in\mathcal{E}} w_{jk}\mathbf{1}(y_j = y_k)}{\sum_{(j,k)\in\mathcal{E}} w_{jk}}, \quad i \in \mathcal{V}, \tag{A.2}$$

where $w_{jk}$ is the weight of the edge between nodes $j$ and $k$.

### B.2.2 Additional Results

The results of homophily ratios of condensed datasets are shown in Table B4. It appears that condensed datasets often struggle to preserve the homophily properties of the original datasets. For instance, in the case of the heterophilous dataset *Flickr*, an increase in the homophily rate is observed under most methods and ratios.

Table B4: Homophily ratio comparison of different condensed datasets

| | Whole Dataset | Ratio ($r$) | GCDM | DM | DosCond | GCond | SGDD |
|---|---|---|---|---|---|---|---|
| *Cora* | 0.81 | 1.30% | 0.76 ↓ | 0.88 ↑ | 0.20 ↓ | 0.64 ↓ | 0.19 ↓ |
| | | 2.60% | 0.11 ↓ | 0.74 ↓ | 0.16 ↓ | 0.55 ↓ | 0.19 ↓ |
| | | 5.20% | 1.00 ↑ | 0.21 ↓ | 0.15 ↓ | 0.62 ↓ | 0.15 ↓ |
| *Citeseer* | 0.74 | 0.90% | 0.16 ↓ | 0.75 ↑ | 0.19 ↓ | 0.57 ↓ | 0.14 ↓ |
| | | 1.80% | 0.08 ↓ | 0.30 ↓ | 0.20 ↓ | 0.36 ↓ | 0.19 ↓ |
| | | 3.60% | 1.00 ↑ | 0.34 ↓ | 0.15 ↓ | 0.22 ↓ | 0.15 ↓ |
| *Flickr* | 0.24 | 0.05% | 0.28 ↑ | 0.29 ↑ | 0.25 ↑ | 0.28 ↑ | 0.32 ↑ |
| | | 0.50% | 0.29 ↑ | 0.22 ↓ | 0.08 ↓ | 0.28 ↑ | 0.30 ↑ |
| | | 1.00% | 0.36 ↑ | 0.18 ↓ | 0.06 ↓ | 0.28 ↑ | 0.26 ↑ |

We visualize the condensed datasets using force-directed graph visualization, as shown in Figure B.1, Figure B.2, and Figure B.3. Since *SFGC* [47] and *GEOM* [45] synthesize edge-free datasets, we do not visualize the datasets they condensed. As shown in the visualization, graphs condensed by different methods exhibit distinct structural characteristics. For example, distribution matching methods often result in less pronounced community structures compared to other methods.

We also visualize the node degree distribution of the original graph and the condesed graphs in Figure B.4. Note that the graphs condensed by *GCDM* [20] and *DM* [24] are dense and each edge has an extremely small weight under most situations, the degree of each node is also small. We observe that the degree distributions of most condensed datasets deviate significantly from the original graph. Among them, *SGDD* [42] demonstrates a relatively similar degree distribution to that of the original graph.

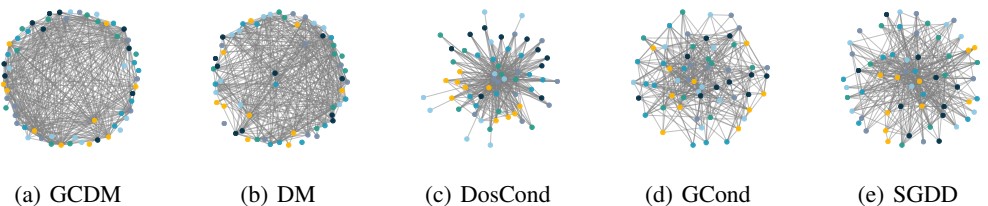

| (a) GCDM | (b) DM | (c) DosCond | (d) GCond | (e) SGDD |

Figure B.1: Visualization of the Condensed *Citeseer* (1.80%) Dataset. Only the top 20% of edges ranked by weight are visualized.

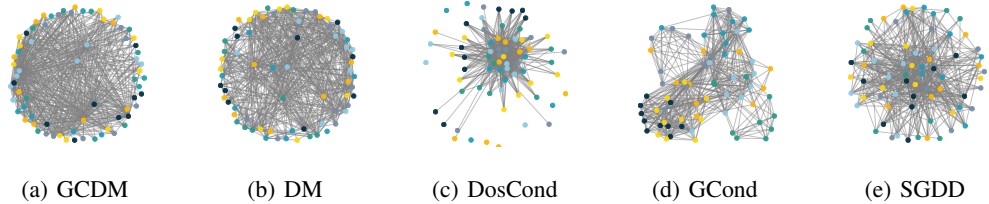

| (a) GCDM | (b) DM | (c) DosCond | (d) GCond | (e) SGDD |

Figure B.2: Visualization of the Condensed *Cora* (2.60%) Dataset. Only the top 20% of edges ranked by weight are visualized.

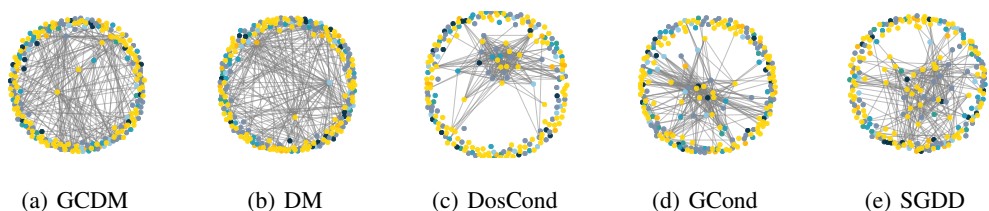

| (a) GCDM | (b) DM | (c) DosCond | (d) GCond | (e) SGDD |

Figure B.3: Visualization of the Condensed *Flickr* (0.50%) Dataset. Only the top 1% of edges ranked by weight are visualized.

## B.3  Settings and Additional Results of Transferability in Different Tasks (RQ3)

### B.3.1  Link Prediction

For the link prediction task, we utilize a graph autoencoder (GAE)[17] based on Graph Convolutional Networks (GCN[16]). The GAE consists of a two-layer GCN encoder that creates node embeddings. During training, we enhance the dataset by randomly adding negative links and use a decoder to perform binary classification on edges. During evaluation, we test the model using the test set of the

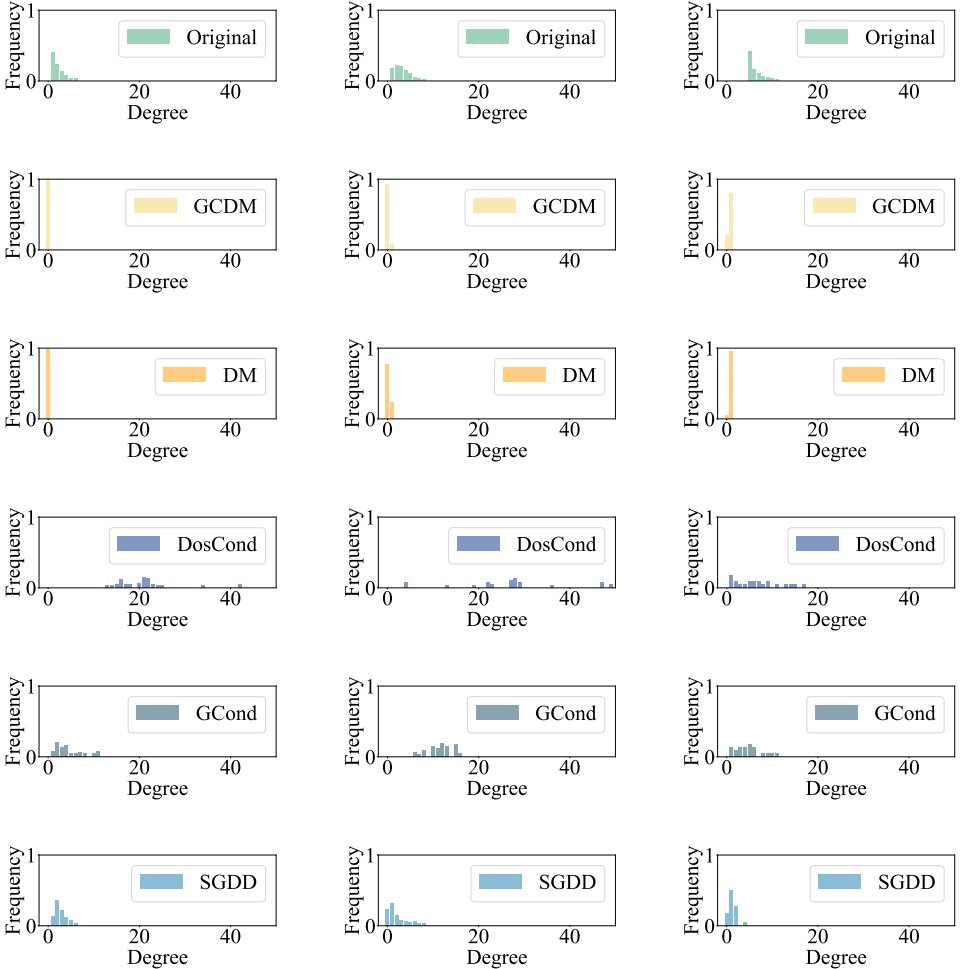

Figure B.4: Degree distribution in the condensed graphs for *Citeseer* (1.80%), *Cora* (2.60%), and *Flickr* (0.05%). The first, second, and third columns correspond to *Citeseer*, *Cora*, and *Flickr*, respectively.

original graph. Since trajectory matching methods do not generate any edges, we do not use them for link prediction tasks. The results of condensed datasets on the link prediction task are shown in Table B5. We observe that most condensed datasets underperform in link prediction tasks, especially on *ogbn-arxiv* and *Flickr*. Most methods' condensed datasets consistently underperform compared to traditional core-set methods, indicating room for improvement.

### B.3.2 Node Clustering

For the node clustering tasks on condensed datasets, we utilize DAEGC [33] to train on synthetic datasets condensed using the node classification task. We then test the trained model on the original large-scale datasets and include the results of other methods on the original graph for comprehensive comparison. Due to the performance degradation of GAT with large neighborhood sizes, we use GCN as the encoder.Performance metrics include Accuracy (Acc.), Normalized Mutual Information (NMI), F-score, and Adjusted Rand Index (ARI).

To fully leverage the condensed datasets, we include the results of node clustering with pertaining. In this experiment, the GCN encoder is first trained on the synthetic datasets with a node classification task, which incorporates the synthetic labels' information. Using the pre-trained GCN as an encoder, we then perform node clustering on the synthetic datasets and the original graph. Results of node clustering tasks, both without and with pertaining are shown in Table B6 and Table B7 respectively.

Table B5: **Link Prediction Accuracy** (%) of different condensed datasets. The best results are shown in **bold**.

| Dataset | Ratio (r) | Random | Herding | K-Center | GCDM | DM | DosCond | GCond | SGDD | Whole Dataset |
|---|---|---|---|---|---|---|---|---|---|---|
| *Citeseer* | 0.90% | 0.52 | 0.52 | 0.55 | 0.53 | 0.53 | 0.50 | 0.65 | **0.69** | |
| | 1.80% | 0.52 | 0.52 | 0.54 | 0.51 | 0.52 | 0.51 | 0.51 | **0.67** | 0.82 |
| | 3.60% | 0.54 | 0.53 | 0.53 | 0.53 | 0.53 | 0.53 | 0.53 | **0.61** | |
| *Cora* | 1.30% | 0.58 | 0.54 | 0.58 | **0.72** | 0.71 | 0.67 | 0.61 | 0.51 | |
| | 2.60% | 0.55 | 0.55 | 0.56 | 0.69 | 0.67 | 0.58 | **0.77** | 0.62 | 0.78 |
| | 5.20% | 0.57 | 0.56 | 0.58 | 0.70 | **0.71** | 0.59 | 0.65 | 0.56 | |
| *ogbn-arxiv* | 0.05% | **0.76** | 0.68 | 0.67 | 0.66 | 0.68 | 0.63 | 0.60 | 0.70 | |
| | 0.20% | 0.72 | 0.72 | **0.73** | 0.72 | 0.72 | 0.69 | 0.71 | 0.51 | 0.75 |
| | 0.50% | **0.74** | 0.73 | **0.74** | 0.71 | 0.73 | 0.72 | 0.72 | 0.70 | |
| *Flickr* | 0.05% | 0.55 | 0.54 | 0.53 | **0.60** | 0.53 | 0.52 | 0.54 | 0.51 | |
| | 0.20% | 0.63 | 0.63 | 0.63 | 0.63 | 0.51 | 0.53 | 0.57 | **0.70** | 0.75 |
| | 0.50% | **0.70** | 0.68 | **0.70** | 0.56 | 0.65 | 0.62 | 0.67 | 0.61 | |

We observe that most condensed datasets perform worse in the node clustering task compared to the original dataset. However, when additional information from the pretraining of the node classification task on condensed dataset is utilized, the results of node clustering significantly improve. Notably, some datasets in Table B6 exhibit identical results with the Adjusted Rand Index (ARI) being 0 or even negative. This occurs because the clustering results do not match the number of classes in the labels, requiring manual splitting of clusters in such scenarios. An ARI of 0 indicates that the clustering result is as good as random, while a negative ARI suggests it is worse than random.

### B.3.3   Anomaly Detection

For the anomaly detection tasks, we generate two types of anomalies, *Contextual Anomalies* and *Structural Anomalies*, following the method described in [3]. We set the anomaly rate to 0.05; if the condensed dataset is too small, we inject one contextual anomaly and two structural anomalies.

**Contextual Anomalies**: Each outlier is generated by randomly selecting a node and substituting its attributes with those from another node with the maximum Euclidean distance in attribute space.

**Structural Anomalies**: Outliers are generated by randomly selecting a small group of nodes and making them fully connected, forming a clique. The nodes in this clique are then regarded as structural outliers. This process is repeated iteratively until a predefined number of cliques are generated.

We conduct anomaly detection by training a DOMINANT model [3], which features a shared graph convolutional encoder, a structure reconstruction decoder, and an attribute reconstruction decoder. Initially, we inject predefined anomalies into the test set of the original graph and use it for evaluation across different condensed datasets derived from this graph. The model is then trained on these condensed datasets, which were injected with specific types of anomalies before training. The DOMINANT model measures reconstruction errors as anomaly scores for both the graph structure and node attributes, combining these scores to detect anomalies. The results are evaluated using the ROC-AUC metric, as shown in Table B8 and B9.

### B.4   Settings and Additional Results of Transferability across Backbone Model Architectures (RQ4)

### B.4.1   Experimental Settings

For transferability evaluation, we use different models as backbones to test the condensation methods. For distribution matching methods, two backbone models with shared parameters are used to generate embeddings that are matched. For trajectory matching methods, two backbone models are used to generate expert trajectories and student trajectories, respectively, to match the corresponding parameters. For gradient matching methods, two backbone models with shared parameters are used to generate gradients for real and synthetic data. Models are selected using grid-searched hyperparameters. The details of the backbone architecture are as follows:

Table B6: **Node Clustering without Pretraining Results** on *Cora* and *Citeseer* with varying condensation ratios ($r$). The best results are highlighted in **bold**, the runner-ups are underlined, and the best results of condensed datasets are shaded in gray .

| Methods | Ratio ($r$) | *Citeseer* | | | | Ratio($r$) | *Cora* | | | |
|---|---|---|---|---|---|---|---|---|---|---|
| | | Acc. | NMI | ARI | F1 | | Acc. | NMI | ARI | F1 |
| K-means | | 54.4 | 31.2 | 28.5 | 41.3 | | 50.0 | 31.7 | 37.6 | 23.9 |
| DAEGC [33] | Full | **67.2** | **39.7** | **41.0** | **63.6** | Full | **70.4** | **52.8** | **68.2** | 49.6 |
| Random | 0.90% | 40.6 | 19.1 | 17.5 | 36.0 | 1.30% | 36.6 | 13.5 | 9.0 | 34.3 |
| | 1.80% | 38.3 | 14.8 | 13.6 | 34.5 | 2.60% | 33.5 | 13.9 | 7.1 | 33.4 |
| | 3.60% | 41.8 | 18.1 | 16.9 | 39.4 | 5.20% | 30.2 | 0.4 | 0.0 | 6.8 |
| Herding | 0.90% | 41.9 | 16.9 | 15.3 | 40.0 | 1.30% | 37.4 | 18.2 | 11.7 | 35.0 |
| | 1.80% | 44.9 | 18.7 | 16.0 | 41.1 | 2.60% | 36.6 | 16.4 | 11.9 | 34.0 |
| | 3.60% | 58.1 | 27.8 | 29.2 | 52.3 | 5.20% | 26.7 | 13.7 | 2.9 | 20.6 |
| K-Center | 0.90% | 37.9 | 13.4 | 11.1 | 35.2 | 1.30% | 34.3 | 13.5 | 7.8 | 32.4 |
| | 1.80% | 50.0 | 23.5 | 22.9 | 46.5 | 2.60% | 42.5 | 22.3 | 15.0 | 42.3 |
| | 3.60% | 31.9 | 14.0 | 10.2 | 31.0 | 5.20% | 30.2 | 0.4 | 0.0 | 6.8 |
| GCDM | 0.90% | 41.4 | 16.9 | 16.2 | 38.6 | 1.30% | 30.2 | 0.4 | 0.0 | 6.8 |
| | 1.80% | 44.1 | 18.1 | 18.1 | 38.8 | 2.60% | 30.2 | 0.4 | 0.0 | 6.8 |
| | 3.60% | 22.8 | 1.8 | 1.2 | 20.9 | 5.20% | 30.2 | 0.4 | 0.0 | 6.8 |
| DM | 0.90% | 23.5 | 2.1 | 1.1 | 17.7 | 1.30% | 30.2 | 0.4 | 0.0 | 6.8 |
| | 1.80% | 45.3 | 19.1 | 17.7 | 42.9 | 2.60% | 29.2 | 2.0 | 0.0 | 9.5 |
| | 3.60% | 25.9 | 4.5 | 3.5 | 20.0 | 5.20% | 30.2 | 0.4 | 0.0 | 6.8 |
| DosCond | 0.90% | 28.6 | 10.2 | 6.3 | 25.1 | 1.30% | 30.2 | 0.4 | 0.0 | 6.8 |
| | 1.80% | 57.1 | 31.4 | 26.2 | 49.5 | 2.60% | 30.2 | 0.4 | 0.0 | 6.8 |
| | 3.60% | 44.3 | 20.6 | 17.0 | 38.6 | 5.20% | 29.6 | 16.2 | 7.7 | 23.4 |
| GCond | 0.90% | 61.8 | 34.0 | 34.7 | 55.9 | 1.30% | 46.6 | 36.7 | 27.3 | 41.2 |
| | 1.80% | 59.6 | 33.0 | 32.6 | 50.3 | 2.60% | 49.9 | 39.3 | 27.9 | 44.3 |
| | 3.60% | 57.8 | 32.0 | 30.2 | 54.8 | 5.20% | 44.6 | 40.9 | 25.1 | 37.3 |
| SGDD | 0.90% | 56.5 | 27.3 | 26.8 | 50.6 | 1.30% | 30.2 | 0.4 | 0.0 | 6.8 |
| | 1.80% | 45.4 | 24.0 | 20.0 | 43.9 | 2.60% | 30.2 | 0.4 | 0.0 | 6.8 |
| | 3.60% | 42.5 | 23.6 | 20.8 | 38.2 | 5.20% | 33.2 | 17.9 | 8.8 | 25.5 |
| SFGC | 0.90% | 46.7 | 19.9 | 18.8 | 43.4 | 1.30% | 42.1 | 23.5 | 17.7 | 39.2 |
| | 1.80% | 56.8 | 27.4 | 27.6 | 52.8 | 2.60% | 54.4 | 31.8 | 26.4 | 50.2 |
| | 3.60% | 47.7 | 19.0 | 16.9 | 45.3 | 5.20% | 30.1 | 0.4 | -0.1 | 6.8 |
| GEOM | 0.90% | 41.4 | 16.9 | 16.2 | 38.6 | 1.30% | 40.7 | 16.9 | 11.6 | 37.3 |
| | 1.80% | 44.1 | 18.1 | 18.1 | 38.8 | 2.60% | 30.8 | 12.9 | 9.3 | 29.2 |
| | 3.60% | 22.8 | 1.8 | 1.2 | 20.9 | 5.20% | 35.6 | 16.0 | 11.5 | 33.6 |

- **MLP:** MLP is a simple neural network consisting of fully connected layers. The MLP we use is structured similarly to a GCN but without the adjacency matrix input, effectively functioning as a standard multi-layer perceptron (MLP). The MLP we adopted consists of 2 layers with 256 hidden units in each layer.

- **GCN [16]:** GCN is the most common architecture for evaluating condensed datasets in mainstream GC methods. GCN defines a localized, first-order approximation of spectral graph convolutions, effectively aggregating and combining features from a node's local neighborhood, leveraging the graph's adjacency matrix to update node representations through multiple layers. We adhere to the setting in previous work [15] and use 2 graph convolutional layers for node classification, each followed by ReLu activation and batch normalization depending on the configuration. For graph classification, we use a 3-layer GCN with a sum pooling function. The hidden unit size is set to 256.

- **SGC [37]:** SGC is the standardized model used for condensation in previous works. It can be regarded as a simplified version of GCN, which ignores the nonlinear activation function but still keeps two Graph Convolution layers, thereby preserving similar graph filtering behaviors. In the experiments, we use 2-layer SGC with no bias.

Table B7: **Node Clustering with Pretraining Results** on *Cora* and *Citeseer* with varying condensation ratios ($r$). The best results are highlighted in **bold** and the runner-ups are underlined.

| Methods | Ratio ($r$) | *Citeseer* | | | | Ratio ($r$) | *Cora* | | | |
|---|---|---|---|---|---|---|---|---|---|---|
| | | Acc. | NMI | ARI | F1 | | Acc. | NMI | ARI | F1 |
| Random | 0.90% | 27.3 | 5.5 | 4.7 | 24.6 | 1.30% | 41.7 | 15.8 | 13.5 | 37.3 |
| | 1.80% | 32.7 | 9.7 | 7.8 | 31.4 | 2.60% | 36.5 | 14.6 | 9.1 | 35.4 |
| | 3.60% | 44.6 | 16.0 | 14.1 | 43.0 | 5.20% | 44.4 | 23.5 | 14.9 | 45.7 |
| Herding | 0.90% | 36.7 | 12.8 | 11.1 | 34.4 | 1.30% | 40.7 | 18.3 | 12.9 | 40.0 |
| | 1.80% | 36.8 | 13.1 | 10.2 | 36.2 | 2.60% | 36.1 | 14.6 | 8.7 | 34.9 |
| | 3.60% | 39.4 | 16.9 | 14.1 | 38.1 | 5.20% | 35.0 | 16.6 | 10.9 | 32.0 |
| K-Center | 0.90% | 33.7 | 9.7 | 8.3 | 29.5 | 1.30% | 41.8 | 19.3 | 14.5 | 39.2 |
| | 1.80% | 37.6 | 15.6 | 13.9 | 34.9 | 2.60% | 38.5 | 20.8 | 14.8 | 38.3 |
| | 3.60% | 41.7 | 17.1 | 14.3 | 40.5 | 5.20% | 38.5 | 17.4 | 10.9 | 36.3 |
| GCDM | 0.90% | 31.1 | 9.6 | 6.6 | 27.3 | 1.30% | 21.3 | 3.7 | 1.7 | 20.1 |
| | 1.80% | 33.1 | 11.9 | 11.1 | 30.4 | 2.60% | 27.0 | 10.9 | 5.7 | 26.7 |
| | 3.60% | 39.7 | 18.0 | 15.2 | 34.4 | 5.20% | 30.0 | 12.4 | 7.0 | 29.6 |
| DM | 0.90% | 36.5 | 15.7 | 12.9 | 30.0 | 1.30% | 27.3 | 9.3 | 4.5 | 25.7 |
| | 1.80% | 37.1 | 10.6 | 8.6 | 31.4 | 2.60% | 20.8 | 3.3 | 0.9 | 19.0 |
| | 3.60% | 29.2 | 6.0 | 4.0 | 23.6 | 5.20% | 23.5 | 4.8 | 1.6 | 16.3 |
| DosCond | 0.90% | **62.7** | **35.9** | **35.1** | **60.6** | 1.30% | 60.2 | 42.5 | 29.4 | 61.2 |
| | 1.80% | 45.2 | 17.9 | 15.4 | 40.8 | 2.60% | 44.5 | 30.1 | 16.6 | 46.5 |
| | 3.60% | 58.6 | 29.6 | 28.5 | 55.8 | 5.20% | 25.4 | 9.8 | 5.0 | 25.0 |
| GCond | 0.90% | 44.0 | 22.5 | 18.7 | 40.3 | 1.30% | 67.4 | 45.1 | 40.4 | 65.8 |
| | 1.80% | 58.5 | 30.9 | 29.6 | 54.9 | 2.60% | 63.7 | 44.5 | 36.2 | 61.8 |
| | 3.60% | 52.0 | 26.8 | 22.5 | 46.6 | 5.20% | 60.9 | 47.1 | 37.1 | 56.0 |
| SGDD | 0.90% | 46.7 | 23.5 | 19.1 | 42.3 | 1.30% | 65.1 | 44.6 | 37.1 | 64.6 |
| | 1.80% | 55.4 | 28.0 | 25.8 | 50.9 | 2.60% | 35.7 | 19.2 | 11.7 | 34.8 |
| | 3.60% | 40.5 | 18.3 | 14.3 | 34.8 | 5.20% | **74.8** | **51.9** | **53.1** | **72.8** |
| SFGC | 0.90% | 34.2 | 9.8 | 8.4 | 32.2 | 1.30% | 41.2 | 21.2 | 13.9 | 40.2 |
| | 1.80% | 47.1 | 21.7 | 20.6 | 43.5 | 2.60% | 38.7 | 20.7 | 13.5 | 36.2 |
| | 3.60% | 48.5 | 23.3 | 21.5 | 44.8 | 5.20% | 37.3 | 21.1 | 14.4 | 34.1 |
| GEOM | 0.90% | 32.7 | 10.5 | 8.6 | 31.7 | 1.30% | 39.1 | 20.1 | 11.4 | 40.0 |
| | 1.80% | 48.2 | 23.6 | 22.7 | 45.2 | 2.60% | 32.2 | 14.5 | 8.9 | 29.4 |
| | 3.60% | 54.2 | 25.7 | 24.9 | 52.1 | 5.20% | 38.1 | 22.0 | 12.7 | 34.7 |

Table B8: **Structural Anomaly Detection results (ROC-AUC)** on *Cora* and *Citeseer* with varying condensation ratios. The best results are shown in **bold** and the runner-ups are shown in underline.

| Dataset | Ratio ($r$) | Random | Herding | K-Center | GCDM | DM | DosCond | GCond | SGDD | SFGC | GEOM |
|---|---|---|---|---|---|---|---|---|---|---|---|
| *Citeseer* | 0.90% | 0.44 | 0.38 | 0.44 | 0.76 | 0.76 | 0.73 | **0.77** | 0.67 | 0.62 | 0.59 |
| | 1.80% | 0.46 | 0.45 | 0.46 | **0.78** | **0.78** | 0.66 | 0.75 | 0.68 | 0.60 | 0.56 |
| | 3.60% | 0.44 | 0.40 | 0.44 | **0.76** | **0.76** | 0.70 | 0.74 | 0.75 | 0.59 | 0.57 |
| *Cora* | 1.30% | 0.56 | 0.59 | 0.62 | 0.80 | 0.80 | 0.79 | **0.81** | 0.75 | 0.54 | 0.51 |
| | 2.60% | 0.50 | 0.65 | 0.67 | 0.80 | 0.80 | **0.82** | 0.79 | 0.81 | 0.53 | 0.53 |
| | 5.20% | 0.65 | 0.55 | 0.67 | **0.82** | **0.82** | **0.82** | 0.81 | 0.71 | 0.54 | 0.55 |

- **Cheby [2]:** Cheby utilizes Chebyshev polynomials to approximate the graph convolution operations, which retains the essential graph filtering properties of GCN while reducing the computational complexity. We use a 2-layer Cheby with 256 hidden units and ReLU activation function.

- **GraphSAGE [12]:** GraphSAGE is a spatial-based graph neural network that directly samples and aggregates features from a node's local neighborhood. In the experiments, We use a two-layer architecture and a hidden dimension size of 256 while using a mean aggregator.

- **APPNP [18]:** APPNP leverages personalized PageRank to propagate information throughout the graph. This method decouples the neural network used for prediction from the propagation mechanism, enabling the use of personalized PageRank for message passing. In the experiments,

Table B9: **Contextual Anomaly Detection results (ROC-AUC)** on *Cora* and *Citeseer* with varying condensation ratios. The best results are shown in **bold** and the runner-ups are shown in underline.

| Dataset | Ratio ($r$) | Random | Herding | K-Center | GCDM | DM | DosCond | GCond | SGDD | SFGC | GEOM |
|---|---|---|---|---|---|---|---|---|---|---|---|
| *Citeseer* | 0.90% | 0.62 | 0.60 | 0.62 | 0.65 | 0.65 | 0.55 | 0.70 | **0.74** | 0.62 | 0.59 |
| | 1.80% | 0.60 | 0.54 | 0.60 | 0.64 | 0.65 | 0.58 | **0.68** | 0.67 | 0.60 | 0.56 |
| | 3.60% | 0.57 | 0.56 | 0.57 | **0.68** | **0.68** | 0.59 | **0.68** | 0.52 | 0.59 | 0.57 |
| *Cora* | 1.30% | 0.52 | 0.48 | 0.53 | 0.52 | 0.52 | 0.45 | **0.54** | 0.41 | **0.54** | 0.51 |
| | 2.60% | 0.50 | 0.45 | 0.54 | 0.54 | 0.54 | 0.56 | 0.55 | **0.57** | 0.53 | 0.53 |
| | 5.20% | 0.56 | 0.58 | 0.59 | 0.55 | 0.55 | 0.55 | 0.57 | **0.62** | 0.54 | 0.55 |

we use a 2-layer model implemented with ReLU activation and sparse dropout to condense and evaluate.

- **GIN [40]:** GIN aggregates features by linearly combining the node features with those of their neighbors, achieving classification power as strong as the Weisfeiler-Lehman graph isomorphism test. We specifically applied a 3-layer GIN with a mean pooling function to compress and evaluate graph classification datasets. For the datasets *DD* and *NCI1*, we use negative log-likelihood loss function for training and softmax activation in the final layer. For *ogbg-molhiv*, *ogbg-molbbbp* and *ogbg-molbace*, we use binary cross-entropy with logits and sigmoid activation in the final layer.

- **Graph Transformer [31]:** The Graph Transformer leverages the self-attention mechanism of the Transformer to capture long-range dependencies between nodes in a graph. It employs multi-head self-attention to dynamically weigh the importance of different nodes, effectively modeling complex relationships within the graph. We use a two-layer model with layer normalization and gated residual connections, following the settings outlined in [31].

### B.4.2 Additional Results

Table B10 shows the node classification accuracy of datasets condensed by traditional core-set methods, which is backbone-free, evaluated across different backbone architectures on *Cora*.

Table B10: **Node Classification Accuracy** (%) of core-set datasets across different backbone architectures on *Cora* (2.6%).

| Methods | SGC | GCN | GraphSage | APPNP | Cheby | GTrans. | MLP |
|---|---|---|---|---|---|---|---|
| Full Dataset | 80.8 | 80.8 | 80.8 | 80.3 | 78.8 | 69.6 | 81.0 |
| Herding | 74.8 | 74.0 | 74.1 | 73.3 | 69.6 | 65.4 | 74.1 |
| K-Center | 72.5 | 72.4 | 71.8 | 71.5 | 63.0 | 64.3 | 72.2 |
| Random | 71.7 | 72.4 | 71.6 | 71.3 | 65.3 | 62.7 | 71.6 |

## B.5 Settings and Additional Results of Initialization Impacts (RQ5)

### B.5.1 Experimental Settings

The details of evaluated initialization mechanism are as follows:

- **Random Sample.** We randomly select features from nodes in the original graph that correspond to the same label, using these features to initialize the synthetic nodes.

- **Random Noise.** Consistent with prevalent dataset condensation methods, we initialize node features by sampling from a Gaussian distribution.

- **Center.** This method involves extracting features from nodes within the same label, applying the K-Means clustering algorithm to these features while treating the graph as a singular cluster and utilizing the centroid of this cluster as the initialization point for all synthetic nodes bearing the same label.

- **K-Center.** Similar to the Center initialization method, but employ the K-Means Clustering method on original nodes by dividing each class of the original graph nodes into n clusters,

where n is the number of synthetic nodes per class. We select the center of these clusters as the initialization of synthetic nodes in this class.

- **K-Means.** Similar to the K-Center initialization method, but instead of using the centroids of clusters to initialize the synthetic dataset, randomly select one node from each cluster to serve as the initial state for the synthetic node.

### B.5.2 Additional Results

The performance of different initialization mechanism on *Cora* (2.6%) and *Cora* (0.26%) are shown in Table B11 and Table B12, respectively. It is evident that distribution matching methods are highly sensitive to the choice of initialization, especially when the dataset is condensed to a smaller scale. Additionally, trajectory matching methods perform poorly with random noise initialization and often fail to converge.

Table B11: Performance comparison of different initialization on various methods for *Cora* (2.60%).The best results are shown in **bold** .

| Methods | Random Noise | Random Sample | Center | K-Center | K-Means |
|---------|--------------|---------------|--------|----------|---------|
| GCDM | 34.5 | 73.3 | 77.4 | **78.7** | 75.9 |
| DM | 34.5 | 73.7 | 77.7 | **78.1** | 75.9 |
| DosCond | 78.8 | 81.9 | 81.8 | **82.5** | 81.8 |
| GCond | 74.8 | 75.1 | **76.3** | 76.2 | 75.1 |
| SGDD | 81.7 | 81.8 | 82.6 | **82.7** | 82.5 |
| SFGC | 52.5 | 80.7 | 79.7 | 81.5 | **81.8** |
| GEOM | - | 77.9 | 48.3 | 78.8 | **78.9** |

Table B12: Performance comparison of different initialization on various methods for *Cora* (0.26%). The best results are shown in **bold** .

| Methods | Random Noise | Random Sample | Center | K-Center | K-Means |
|---------|--------------|---------------|--------|----------|---------|
| GCDM | 32.3 | 37.8 | **78.7** | **78.7** | 34.3 |
| DM | 32.2 | 38.4 | **77.9** | **77.9** | 34.2 |
| DosCond | 78.7 | **82.4** | 80.5 | 82.0 | 81.9 |
| GCond | 80.2 | **81.6** | 80.1 | 81.2 | 80.7 |
| SGDD | 82.2 | 82.2 | **82.7** | **82.7** | 81.5 |
| SFGC | 79.7 | 79.7 | **79.8** | **79.8** | 72.0 |
| GEOM | - | 49.6 | 51.3 | 51.3 | **65.0** |

## B.6 Settings and Additional Results of Efficiency and Scalability (RQ6)

### B.6.1 Experimental Settings

For a fair comparison, all the experiments are conducted on a single NVIDIA A100 GPU. Then we report the overall condensation time (min) when achieving the best validation performance, the peak CPU memory usage (MB) and the peak GPU memory usage (MB).

### B.6.2 Additional Results

The detailed time and space consumption of the node-level GC methods on *ogbn-arxiv* (0.50%) and graph-level GC methods on *ogbg-molhiv* (1 Graph/Cls) are shown in Table B13 and Table B14 respectively. For node-level methods, although trajectory matching methods (*SFGC* [47], *GEOM* [45]) may consume less time and memory due to their offline matching mechanism, the expert trajectories generated before matching can occupy up to 764 GB of space as shown in Table B15, significantly impacting storage requirements. Among all the graph-level GC methods, *Mirage* [11] stands out by not relying on any GPU resources for calculation and can condense data extremely quickly, taking only 1% of the time required by other methods.

Table B13: Time and memory consumption of different methods on *ogbn-arxiv* (0.50%).

| Consumption | GCDM | DM | DosCond | GCond | SGDD | SFGC | GEOM |
|-------------|------|------|---------|-------|------|------|------|
| Time (min) | 212.90 | 57.70 | 117.38 | 266.57 | 226.62 | 245.65 | 148.37 |
| Acc. (%) | 58.09 | 58.09 | 60.73 | 61.28 | 61.51 | 67.13 | 67.29 |
| CPU Memory (MB) | 2720.88 | 2708.70 | 5372.60 | 5270.70 | 5426.30 | 3075.30 | 3335.10 |
| GPU Memory (MB) | 2719.74 | 2552.63 | 3850.24 | 3850.24 | 8326.35 | 4050.12 | 5308.42 |

## C  Reproducibility and Limitations

**Accessibility and license.** All the datasets, algorithm implementations, and experimental settings are publicly available in our open project (`https://github.com/RingBDStack/GC-Bench`). Our

Table B14: Time and memory consumption of different methods on *ogbg-molhiv* (1 Graph/Cls).

| Consumption | DosCond | KiDD | Mirage |
|---|---|---|---|
| Time (min) | 218.11 | 202.38 | 2.91 |
| Acc. (%) | 67.41 | 66.44 | 71.09 |
| CPU Memory (MB) | 2666.29 | 3660.79 | 752.22 |
| GPU Memory (MB) | 1005.98 | 6776.42 | 0.00 |

Table B15: Expert trajectory size (GB) for trajectory matching methods.

| *Citeseer* | *Cora* | *ogbn-arxiv* | |
|---|---|---|---|
| 129 | 152 | 15 | |
| *Flickr* | *Reddit* | *ACM* | *DBLP* |
| 21 | 42 | 312 | 764 |

package (codes and datasets) is licensed under the MIT License. This license permits users to freely use, copy, modify, merge, publish, distribute, sublicense, and sell copies of the software, provided that the original copyright notice and permission notice are included in all copies or or substantial portions of the software. The MIT License is widely accepted for its simplicity and permissive terms, ensuring ease of use and contribution to the codes and datasets. We bear all responsibility in case of violation of rights, *etc*, and confirmation of the data license.

**Datasets.** *Cora*, *Citeseer*, *Flickr*, *Reddit* and *DBLP* are publicly available online[3] with the MIT license. *ogbn-arxiv*, *ogbg-molbace*, *ogbg-molbbbp* and *ogbg-molhiv* are released by OGB [13] with the MIT license. *ACM* [46] is the subset hosted in [35] with the MIT license. *NCI1* [32] and *DD* [4] are available in TU Datasets [27] with the MIT license. All the datasets are consented to by the authors for academic usage. All the datasets do not contain personally identifiable information or offensive content.

**Limitations.** GC-Bench has some limitations that we aim to address in future work. Our current benchmark is limited to a specific set of graph types and graph tasks and might not reflect the full potential and versatility of GC methods. We hope to implement more GC algorithms for various tasks (e.g. subgraph classification, community detection) on more types of graphs (e.g., dynamic graph, directed graph). Besides, due to resource constraints and the availability of implementations, we could not include some of the latest GC algorithms in our benchmark. We will continuously update our repository to keep track of the latest advances in the field. We are also open to any suggestions and contributions that will improve the usability and effectiveness of our benchmark, ensuring it remains a valuable resource for the IGL research community.

---

[3]https://github.com/pyg-team/pytorch_geometric

