# OpenReview forum: "GC-Bench: An Open and Unified Benchmark for Graph Condensation"
_NeurIPS.cc/2024/Datasets_and_Benchmarks_Track — NeurIPS 2024 Track Datasets and Benchmarks Poster_

### Official Review · Reviewer_qXuL · 2024-07-01
**Review for GC-Bench**

**Rating:** 8
**Confidence:** 3
**Correctness:** The benchmark is constructed appropri…
**Clarity:** This paper is well-organized.

**Review:**

**Pros:**
1. Graph condensation has attracted significant research interest in the field in recent years. This paper effectively summarizes and benchmarks existing graph condensation algorithms, which could greatly benefit the research community. It facilitates quickly identifying under-explored points while saving time reproducing existing methods.
2. The evaluation dimensions and the conducted experiments are thoroughly comprehensive. The released code repository is well-documented and ready for use.
3. The selected categories of algorithms are representative, and the chosen tasks are insightful, for instance, the structure and initialization discussions.
4. The proposed takeaways are intriguing and worth further investigating. I believe they may prompt the graph condensation community.
5. The investigation of the transferability of tasks and backbones is also insightful. The exploration of cross-task problems is an inspiring direction.

**Cons:**
1. The analysis of the experimental results should be enhanced, which will be helpful for more insightful findings.
2. The discussion concerning the structure, particularly for heterogeneous graphs, requires more distinctive experiments.
3. The description of Figure 3(c) and Figure 4 should be enhanced, the present results are not evident.

**Strengths:**

1. The proposed GC-Bench investigates the graph condensation methods, including the gradient, trajectory, distribution, kernel ridge regression, and computation tree compression methods. It also conducts both node-level and graph-level experiments on 12 diverse graph datasets.
2. The challenges and insights proposed by GC-Bench are both novel and significant. For instance, the heterogeneity and heterophily properties are critical to graph neural networks yet remain largely unexplored by current researchers. The counterintuitive phenomenon of the graph transformer in the cross-architecture also raises my interest, suggesting that it may need further investigation by the community.
3. The new problem raised about the cross-task applications is substantial. It is intriguing how the task-oriented graph condensation can retain the task-agnostic information, and how it can benefit the downstream models. I believe this could also be a good direction for the following works.

**Additional Feedback:**

See above.

**Documentation:**

This paper provides detailed illustration for benchmark design and evaluation settings.

**Limitations:**

The authors provide discussion for the limitations.

**Opportunities For Improvement:**

1. It would be beneficial to provide a more in-depth analysis of the impact of structure on graph condensation. While I agree that the two properties discussed can describe the structure from the perspective of graph neural networks, it would enhance the paper to discuss additional aspects of graph structure.
2. Although GC-Bench conducts experiments with heterogeneous graphs, the choice of GNN backbones is still limited to GCN. It would be more informative to compare these results with state-of-the-art heterogeneous GNN to further investigate the insights.
3. For Figure 4, it's challenging to discern the color differences between K-Center and Center. It looks like that K-Center is always the most efficient initialization method? Perhaps changing the colors could better highlight the differences. Additionally, I'm curious about the varying effectiveness across different datasets, which intuitively might be influenced by their scale. Could you provide more results on other datasets and detail their scales?

**Relation To Prior Work:**

The authors clearly discussed in main content and appendix how their work is different from previous contributions.

**Summary And Contributions:**

The provided excerpt outlines the emergence of Graph Condensation (GC) as a prominent method for reducing large-scale graph datasets while maintaining their fundamental properties. Despite the proliferation of GC methods, a comprehensive evaluation and in-depth analysis are lacking, hindering the understanding of advancements in this domain. To address this gap, the authors introduce the Graph Condensation Benchmark (GC-Bench), aiming to systematically analyze the performance of graph condensation across various scenarios.

The GC-Bench evaluates 12 state-of-the-art graph condensation algorithms in both node-level and graph-level tasks, considering characteristics such as effectiveness, transferability, and complexity. This evaluation spans 12 diverse graph datasets, providing a comprehensive understanding of the methods’ performance. Additionally, the authors have developed an accessible library for training and evaluating different GC methods, promoting reproducible research in this area.

---

> ### Author Rebuttal · Authors · 2024-08-17
>
> Dear Reviewer qXuL,
>
> We appreciate your recognition of our paper and agree that it has the potential to prompt the community! Our response to your questions are as follow:
>
> > C1: The analysis of the experimental results should be enhanced, which will be helpful for more insightful findings.
> >
>
> A1: Thanks for your suggestions. We have added the experiments on heterogeneous recommendation data, the impact of structural properties, the fairness comparison, and the converge analysis in the rebuttal phase. We will continue to provide more comprehensive experiments and more detailed and in-depth analyses to improve the revised manuscript.
>
> > C2: The discussion concerning the structure, particularly for heterogeneous graphs, requires more distinctive experiments.
> >
>
> A2: Current graph condensation methods do not support the direct adoption of heterogeneous Graph Neural Networks. Since the primary focus of this benchmark is to provide a more comprehensive understanding and limitation discussion of current GC methods, we simply adopt GCN for comparison. To help better understand the performance of existing graph condensation methods on heterogeneous graph datasets, we have added the SOTA results on these datasets, as shown in the table below. We will include the evaluation with the heterogeneous graph neural networks in the future.
>
> |  | HAN | HGT | MAGNN | GTN | RGCN |
> | --- | --- | --- | --- | --- | --- |
> | ACM | 92.21 | 92.73 | 89.33 | 91.97 | 91.59 |
> | DBLP | 93.73 | 93.08 | 93.70 | 94.51 | 92.29 |
>
> > C3: The description of Figure 3(c) and Figure 4 should be enhanced. The color differences between K-Center and Center. in Figure 4. The varying effectiveness across different datasets scale.
> >
>
> A3: Thank you for your feedback. We will enhance the descriptions of Figure 3(c) and Figure 4 in the revised manuscript to make the results more evident. Regarding the color differences in Figure 4, we recognize the challenge and have updated the rebuttal PDF with more distinctive colors to better differentiate the lines. It’s important to note that K-Center is not always the most efficient initialization method—while it works better for DosCond, Center sometimes performs better on Cora in GCond (as mentioned in **lines 238-240**). We will update the figure in the manuscript accordingly and clarify that the effectiveness depends on the specific dataset and method used.
>
> Regarding the impact of dataset scale on performance, it is indeed a significant factor. Larger datasets, such as ogbn-arxiv and Reddit, generally struggle to achieve lossless condensation, unlike smaller datasets. Below, we provide results that correlate dataset scale with condensation performance. We used similar ratios across four datasets of varying scales, averaged the performance across different methods, and reported the **average performance relative to the whole dataset**:
>
> | Dataset | Scale (#Nodes) | Ratio | Distribution Matching | Gradient Matching | Trajectory Matching |
> | --- | --- | --- | --- | --- | --- |
> | Citeseer | 3,327 | 0.36% | 55.44% | 100.79% | 98.12% |
> | Flickr | 89,250 | 0.50% | 104.38% | 96.30% | 100.00% |
> | ogbn-arxiv | 169,343 | 0.50% | 83.01% | 87.79% | 93.92% |
> | Reddit | 232,965 | 0.50% | 90.74% | 81.76% | 97.55% |
>
> It will be interesting to explore condensation on extremely large-scale datasets (such as Freebase in heterogeneous datasets) in future work, though time constraints prevent us from doing so in the current study, we recognize the importance of this direction and will pursue it in subsequent research.

---

> > ### Comment · Reviewer_qXuL · 2024-08-20
> > **Response to the Authors**
> >
> > Thanks for the authors' detailed response, which has addressed my concerns. I raise the score to 8 to support this work.

---

### Official Review · Reviewer_9zaP · 2024-07-20
**Review of Submission167**

**Rating:** 6
**Confidence:** 4
**Correctness:** Yes.
**Clarity:** Yes.

**Review:**

Please refer to "Strengths" and "Opportunities For Improvement" for details.

**Strengths:**

1.	The paper introduces GC-Bench as a thorough and systematic benchmarking tool that evaluates a wide range of graph condensation methods across various dimensions.
2.	The authors have made the GC-Bench library publicly available, which promotes transparency and reproducibility in research.
3.	The paper is well written and easy to follow.

**Additional Feedback:**

N/A

**Documentation:**

Yes.

**Ethics:**

No.

**Limitations:**

Yes.

**Opportunities For Improvement:**

1.	Why is it that low condensation ratio can outperform the accuracy of the whole dataset for node classification? Yet higher condensation rates instead lead to lower accuracy? More analysis should be provided.
2.	I would like to know the accuracy of graph transformer on the whole dataset in Section 3.4, which may help explain why graph transformer-based methods perform poorly.
3.	As far as I know, DosCond, KiDD and other methods are difficult to converge during model training. It would be better to have a convergence analysis.
4.	It is necessary to evaluate the performance of condensed graphs in practical applications, such as for training and tuning a GNN model.
5.	I disagree with the 258-263 lines’ viewpoint. Although the current method is not efficient and scalable, the condensed graph can facilitate subsequent training. The purpose of graph condensation is to generate a synthetic graph instead of an original graph for training, thus avoiding training on a large-scale graph.
6.	The evaluation criteria should include the fairness of the synthesized dataset. [1,2] observed group fairness issues with graphs synthesized by existing methods.
[1] Feng Q, Jiang Z S, Li R, et al. Fair graph distillation[J]. Advances in Neural Information Processing Systems, 2024, 36.
[2] Mao R, Fan W, Li Q. Gcare: Mitigating subgroup unfairness in graph condensation through adversarial regularization[J]. Applied Sciences, 2023, 13(16): 9166.

**Relation To Prior Work:**

Yes.

**Summary And Contributions:**

It’s a comprehensive paper that includes node-level and graph-level condensation methods. This paper presents several important observations, such as the current limitations of graph-level condensation methods, the impact of dataset characteristics on condensation performance, and the challenges of transferring condensed datasets to different tasks or backbone architectures. In addition to an open-sourced benchmark library, facilitating reproducible research and future exploration in the field of graph condensation.

---

> ### Author Rebuttal · Authors · 2024-08-17
>
> > C5: The 258-263 lines’ viewpoint. Although the current method is not efficient and scalable, the condensed graph can facilitate subsequent training.
> >
>
> A5: We agree that the primary goal of graph condensation is to create a synthetic graph that reduces the computational burden for subsequent training. Our intention was to highlight that the current gradient matching and distribution matching GC methods can lead to Out of Memory (OOM) errors on large-scale datasets with high condensation ratios. This limitation challenges the practicality of these methods for condensing large-scale graphs, which we acknowledge may not have been clearly communicated.
>
> To address this, we will revise the manuscript to better reflect our intended message:
>
> *However, as shown in Table 2, current gradient matching and distribution matching GC methods may result in OOM (Out of Memory) errors on large-scale datasets when high condensation ratios are applied. **These methods cannot be applied to large-scale scenarios, which contradicts the original intent of graph condensation to be employed in extremely large-scale graphs.***
>
> We hope this revision clarifies our point, and we appreciate your understanding.
>
> > C6: The evaluation criteria should include the fairness of the synthesized dataset.
> >
>
> A6: We add two fairness evaluation in the benchmark evaluator according to the setting in GCARe[C] and FGD[D]. For the degree bias, the nodes in datasets are divided into three subgroups according to the thresholds of the degrees. (For `Cora` the testing nodes are divided into subgroups of size `(400, 248, 252)`) $\Delta$acc was set as the accuracy gap between the most and least advantaged subgroups while $\sigma$acc was set as the standard deviation of accuracies across all groups.
>
> For the attribute biases, a binary sensitive attribute is used to split the subgroups as in unified[E]. (For `Credit` the attribute is Age) We use Statistical parity and Equal opprtunity to evaluate the statistical dependencies between predictions and attributes as in fairness[F].
>
> The results of `Cora` and `Credit` are shown in the following table. It can be observed that for degree biases, SFGC and GEOM has comparatively better performance, indicating the possibility of condensation methods not encoding the degree biases using trajectory matching.
>
> We already include the related code in the repository and will include the results in the final revised manuscript.
>
> | dataset | Metric | Random | K-Center | Herding | GCDM | DM | DosCond | GCond | SGDD | SFGC | GEOM | Whole |
> | --- | --- | --- | --- | --- | --- | --- | --- | --- | --- | --- | --- | --- |
> | Cora(2.6%) | $\downarrow \Delta$acc | 13.72 | 10.81 | 9.34 | 11.59 | 10.78 | 8.40 | 11.00 | 12.45 | 7.75 | 8.16 | 8.84 |
> |  | $\downarrow \sigma$acc | 6.96 | 5.47 | 5.33 | 5.79 | 5.41 | 4.51 | 5.64 | 6.95 | 4.43 | 4.63 | 4.55 |
> | Credit(5%) | $\uparrow$ACC | 71.94 | 76.55 | 71.60 | 76.02 | 74.93 | 74.06 | 74.17 | 72.34 | 73.99 | 77.86 | 77.13 |
> |  | $\downarrow \Delta$SP | 7.57 | 2.48 | 9.40 | 5.23 | 1.37 | 7.62 | 8.49 | 12.69 | 4.67 | 3.36 | 6.11 |
> |  | $\downarrow \sigma$EO | 5.79 | 1.73 | 6.60 | 3.73 | 1.16 | 5.74 | 5.75 | 11.85 | 4.12 | 2.99 | 4.36 |
>
> [C] GCARe: Mitigating Subgroup Unfairness in Graph Condensation through Adversarial Regularization, Mao et.al, App.Science.
>
> [D] Fair Graph Distillation, Feng et.al, NeurIPS 2023.
>
> [E] Towards a Unified Framework for Fair and Stable Graph Representation Learning. Chirag, UAI 2021.
> [F] A Comprehensive Survey on Trustworthy Graph Neural Networks: Privacy, Robustness, Fairness, and Explainability, [Enyan](https://dblp.org/pid/250/2886.html), arXiv 2022.

---

> ### Author Rebuttal · Authors · 2024-08-17
>
> Dear Reviewer 9zaP,
>
> Thanks for your constructive suggestions, we provide the responses as follow:
>
> > C1: Why is it that low condensation ratio can outperform the accuracy of the whole dataset for node classification?
> >
>
> A1: Thank you for raising this point. While it may seem counter-intuitive that low condensation ratios can outperform the accuracy of the entire dataset, we believe this can be explained by the condensation methods' ability to filter out noise and remove outliers from the graph’s training set, preserving only the most informative and valuable parts during the distillation process. This phenomenon has also been observed in previous works such as KiDD [A] and GEOM [B].
>
> For higher condensation ratios, as mentioned in **Line 174**, the larger synthetic datasets may introduce biases and noise that can degrade performance. Additionally, as noted in **Line 224**, the extensive matching steps involved in the condensation process might encode model-specific biases into the datasets, leading to worse performance.
>
> [A] Kernel Ridge Regression-Based Graph Dataset Distillation, Xu [et.al](http://et.al), KDD 2023.
>
> [B] Navigating Complexity: Toward Lossless Graph Condensation via Expanding
> Window Matching, Zhang et.al, ICML 2024.
>
> > C2: I would like to know the accuracy of graph transformer on the whole dataset in Section 3.4, which may help explain why graph transformer-based methods perform poorly.
> >
>
> A2: Thank you for your inquiry. The accuracy of the graph transformer on the entire Cora dataset is 76.3%. We believe the poor performance of graph transformer-based methods when applied to condensed graphs may be attributed to the fact that GNN-based methods often struggle when transferred to non-GNN architectures. This is likely due to current condensation methods encoding model-specific information in the synthetic graph, as mentioned in **Line 224**.
>
> To further illustrate this, we provide the results of condensed Citeseer (0.90%) on the graph transformer using different methods below:
>
> | Random | K-Center | Herding | GCDM | DM | DosCond | GCond | SGDD | SFGC | GEOM | Whole |
> | --- | --- | --- | --- | --- | --- | --- | --- | --- | --- | --- |
> | 50.70% | 51.00% | 50.20% | 47.30% | 50.30% | 49.90% | 47.90% | 51.30% | 51.30% | 48.70% | 66.7% |
>
> > **C3:** As far as I know, DosCond, KiDD and other methods are difficult to converge during model training. It would be better to have a convergence analysis.
> >
>
> **A3:** Thank you for the insightful suggestion. Convergence analysis is indeed a valuable perspective that can provide a more comprehensive understanding of existing graph condensation methods. We have already presented the convergence performance of GCond in **Figure 4** of the submitted manuscript. Additionally, we have included further convergence analysis for DosCond and KiDD in the attached rebuttal PDF and will incorporate more detailed analysis in the final version of the manuscript.
>
> Regarding KiDD, it is true that convergence can be challenging. For smaller graphs per classes, KiDD typically converges within 15 epochs as discussed in KiDD[A], whereas larger graphs or more complex class settings require longer training times.
>
> As for DosCond, its convergence speed can be significantly improved with proper initialization, such as using K-Center initialization, as mentioned in **lines 243-245** of our manuscript.
>
> We appreciate your suggestion and hope the additional analysis addresses your concerns.
>
> > C4: It is necessary to evaluate the performance of condensed graphs in practical applications, such as for training and tuning a GNN model.
> >
>
> A4: We agree that evaluating graph condensation methods in more practical applications is highly valuable. In our submitted manuscript, we focused on evaluating cross-task and cross-architecture transferability, which are scenarios commonly encountered in practical applications. To further address this, we have also conducted experiments on the application of Neural Architecture Search (NAS), a primary real-world use case for condensation methods.
>
> In this experiment, we used APPNP as the architecture and explored various configurations, including the number of layers (ranging from 2 to 6), residual coefficients (ranging from 0.1 to 0.2), hidden dimensions (ranging from 8 to 256), and activation functions (including Sigmoid, Tanh, ReLU, Linear, and LeakyReLU). We evaluated the performance of models trained on Cora condensed to 2.6% by different methods across 300 architectures over 10 runs. We report the Pearson correlation between the validation accuracy on the original graph and the condensed graph.
>
> Our results indicate that while distribution methods and traditional coreset methods may achieve high correlation between the ranking of models trained on the condensed and original datasets, they tend to have lower test accuracy on the best-searched architecture. On the other hand, gradient matching and trajectory matching methods can achieve high performance on the best-searched model, but they exhibit lower ranking correlation, making it challenging to reflect the relative strengths of various architectures.
>
> We will include these findings in the final manuscript to provide a more comprehensive evaluation of the practical applicability of graph condensation methods.
>
> |  | Pearson Correlation | Test Accuracy |
> | --- | --- | --- |
> | Herding | 0.72 | 77.5 |
> | K-Center | 0.73 | 73.8 |
> | GCDM | 0.69 | 71.7 |
> | DM | 0.73 | 73.5 |
> | DosCond | 0.42 | 79.8 |
> | GCond | 0.48 | 81.1 |
> | SGDD | 0.66 | 82.0 |
> | SFGC | 0.68 | 80.1 |
> | GEOM | 0.68 | 80.5 |
> | Whole | - | 82.6 |

---

### Official Review · Reviewer_GyyU · 2024-07-24
**Good idea, but execution lacks details**

**Rating:** 6
**Confidence:** 4

**Review:**

This is a timely contribution, and I appreciate the depth of the proposed experiments. There are some issues with the current paper that would need to be addressed, though:

- The paper states that datasets are 'adapted' to this context, but I do not find any details about this. It seems to me that the any graph-learning dataset can be used for this benchmark and the described datasets are not specifically pre- or postprocessed. Is this correct?
- I am missing a simple feature-based baseline for the condensation, i.e. a baseline algorithm that does not necessarily make use of the graph structure. This would be highly important to understand to what extent the predictive performance of a condensed graph is driven more by the presence of certain nodes, rather than the presence of certain structural information of the graph. It would strengthen the proposed framework if such a baseline could be provided (maybe the code for this is already in place and I missed it!)
- There is no versioning concept proposed for the datasets, and the installation instructions mention that one needs to download the datasets manually. This is not a problem per se, but in light of the manuscript claiming that datasets have been specifically selected for condensation purposes, this seems a little bit out of place. If _any_ graph-learning dataset can be used (within reason), I would suggest to rewrite the paper accordingly.
- Apart from that, the proposed software package could benefit from a simpler interface. Is it possible to include some existing data loaders, for instance from `pytorch-geometric`, in order to simplify the setup process?

**Strengths:**

I appreciate the comprehensive experiments and detailed research questions. Moreover, the experimental setup appears to be thorough, encompassing a strong hyperparameter search.

**Additional Feedback:**

Some additional feedback for improving the text: In the introduction, the differences between 'coarsening, sparsification, and condensation' should be described more clearly. The sentence in l. 25 is not providing a good explanation in its current form. Likewise, the discussion could be strengthened by substantiating statements like 'It's not necessarily true that larger compressed datasets lead to better results' with more examples and numbers.

**Clarity:**

The paper is well-written, there are only some semantic issues that should be addressed (see above).

**Correctness:**

The claims in the submission are correct and all benchmarks/experiments are performed in a sound manner.

**Documentation:**

The documentation could be improved with more examples, a specific API documentation, and better instructions (or auxiliary scripts) for dealing with datasets.

**Ethics:**

There are no specific ethical concerns pertaining to this submission.

**Limitations:**

Limitations are only partially addressed. While I understand the limitations in terms of graph types (for instance), I would appreciate a more detailed discussion on data limitations as such, in particular in light of dataset quality. Since the submission is making use of existing datasets, existing quality issues will _not_ be necessarily addressed. This is fine but it needs a better disclaimer and explanation.

**Opportunities For Improvement:**

Given the large number of detailed tables, a [critical difference plot](https://github.com/mirkobunse/critdd) would be highly useful for showing the differences (or lack thereof) of different algorithms.

**Relation To Prior Work:**

Prior work is discussed adequately.

**Summary And Contributions:**

This submission provides a framework for benchmarking graph-condensation algorithms. To this end, 12 datasets and 12 graph-condensation algorithms are systematically evaluated, following a number of guiding research questions concerning (a) overall progress, (b) structural information, (c) transferability, (d) choice of architecture, (e) choice of initialisation strategy, (f) computational efficiency. The main thrust of the submission lies in facilitating the running of experiments on different models to assess the performance under these facets.

---

> ### Author Rebuttal · Authors · 2024-08-17
>
> > C3: If *any* graph-learning dataset can be used (within reason), I would suggest to rewrite the paper accordingly.
> >
>
> A3: Thank you for the valuable suggestion. We will improve the documentation in our repository to better clarify the supported data formats, and we plan to introduce more automated features for downloading and processing datasets. Currently, we support most of the datasets available in PyTorch-Geometric.
>
> It’s important to note that our benchmark specifically addresses node classification and graph classification tasks, which align with the focus of the majority of existing condensation methods, as detailed in Appendix A.2, Table A2. All the algorithms, except for SGDD, are designed for datasets used in these tasks. While it is theoretically possible to adapt these methods to other datasets, we aim to implement them in the benchmark with minimal modifications, particularly for datasets that are tailored to tasks other than Node Classification and Graph Classification.
>
> > C4: Apart from that, the proposed software package could benefit from a simpler interface.
> >
>
> A4: Thank you for the suggestion. We already have this functionality in place. Every `pytorch-geometric` dataset (as long as it can be used for node classification and graph classification, including both heterogeneous and homogeneous datasets) is fully supported. Users can simply utilize the `PyG` dataloader, and the benchmark will handle the rest. Additionally, users can specify any additional processing steps, such as sparsification, as needed.
>
> > C5: Given the large number of detailed tables, a [critical difference plot](https://github.com/mirkobunse/critdd) would be highly useful for showing the differences (or lack thereof) of different algorithms.
> >
>
> A5: We include a critical difference plot in the **Figure A of the general respnse PDF file** to better illustrate the differences and similarity between the various algorithms.
>
> > **C6:** Detailed discussion on data limitations as such, in particular in light of dataset quality.
> >
>
> **A6:** Thank you for highlighting this important aspect. For graph datasets, low quality datasets generally fall into two categories: 1) imbalanced datasets and 2) noisy real-world datasets.
>
> We add several imbalanced datasets for evaluation, as shown below, where $\rho$  represents the imbalance ratio (the ratio between the number of samples in the head and tail classes), and L0-L6 represent the label distributions:
>
> | Dataset | $\rho$ | L0 | L1 | L2 | L3 | L4 | L5 | L6 |
> | --- | --- | --- | --- | --- | --- | --- | --- | --- |
> | Cora | 5 | 30.21 | 15.73 | 15.43 | 12.96 | 11.00 | 8.01 | 6.65 |
> | Citeseer | 3 | 21.07 | 20.08 | 17.91 | 17.73 | 15.26 | 7.94 | - |
> | Flickr | 10 | 40.26 | 25.73 | 9.53 | 7.19 | 5.90 | 5.49 | 3.90 |
>
> Regarding noisy real-world datasets, while our current submission doesn’t include explicit noise handling, we can simulate noise by adding it to features or adjacency matrices if needed. We would be happy to provide additional results during the discussion phase.
>
> We will also ensure that the final manuscript includes a more detailed discussion and disclaimer regarding these data limitations.
>
> > C7: Some additional feedback for improving the text:
> >
>
> A7: Thanks for the suggestions, we would revise the manuscript to :
>
> - Line 25: Compared to former graph reduction methods like graph coarsening and sparsification, graph sparsification generally focuses on approximating a graph by selecting a subset of edges, while graph coarsening groups nodes into "super nodes." In contrast, the investigated graph condensation aims to synthesize a smaller yet highly informative graph that still allows the model to achieve strong performance, comparable to using the full dataset.
> - Lines 172-174: As shown in Table 2, SFGC's performance on the Citeseer dataset decreases from an accuracy of 71.0% at a condensation ratio of 2.7% to 70.6% at a ratio of 3.6%. Similar trends can be observed with GCond on Cora, SGDD on Reddit, DosCond on ogbn-arxiv, and so on. Such trend reveal that a large condensation ratio does not necessarily lead to better performance with current methods.

---

> > ### Comment · Reviewer_GyyU · 2024-08-30
> >
> > Thanks for engaging with my concerns! I will take this into account as we move into the discussion phase.

---

> ### Author Rebuttal · Authors · 2024-08-17
>
> Dear reviewer GyyU,
>
> Thanks for your time and effort on reviewing our work, and glad to see that you recognize our paper as timely contribution. Our response are as follow:
>
> > **C1:** The paper states that datasets are "adapted" to the context but doesn't specify how.
> >
>
> A1: Thank you for raising this point. Not all graph-learning datasets are suitable for this benchmark. Our focus is on datasets used for node classification and graph classification tasks, which align with the core focus of existing graph condensation work. Currently, the benchmark supports most PyG datasets (including both homogeneous and heterogeneous graphs), datasets from GraphSAINT [A] (such as ogbn-arxiv, Reddit, and Flickr), and datasets from HGB [B] (such as ACM and DBLP). We are committed to continuously updating our repository to support more dataset formats.
>
> Additionally, we have specific preprocessing and postprocessing steps for the datasets in the benchmark. Users can specify preprocessing options like feature normalization, and during the evaluation phase, postprocessing steps such as adjacency matrix sparsification can be applied (e.g., for Cora and Citeseer, the edge truncation threshold is set to 0.05 by default, and for others, it is 0.01, with customizable thresholds). We have refined the documentation to provide clearer explanations of these processes.
>
> [A] GraphSAINT: Graph Sampling Based Inductive Learning Method, HangQing et.al, arXiv 2019
>
> [B] Are we really making much progress? Revisiting, benchmarking, and refining heterogeneous graph neural networks, QingSong et.al, KDD 2021.
>
> > C2: I am missing a simple feature-based baseline for the condensation, i.e. a baseline algorithm that does not necessarily make use of the graph structure.
> >
>
> A2: Thank you for the suggestion. We implemented the feature-only baseline, which is equivalent to using an MLP backbone with an identity matrix as the adjacency matrix, effectively removing structural information. However, due to space constraints, we did not display these results in the submitted manuscript.
>
> Here are the results for the MLP-backbone baseline on Cora with a 2.6% condensation ratio:
>
> | Backbone | GCDM | DM | DosCond | GCond | SGDD | SFGC | GEOM |
> | --- | --- | --- | --- | --- | --- | --- | --- |
> | MLP | 74.17 | 74.23 | 76.86 | 74.70 | 72.77 | 72.69 | 69.01 |
> | SGC | 73.4 | 74.62 | 79.62 | 80.51 | 81.82 | 81.72 | 76.14 |
>
> These results show that the feature-only methods using the MLP backbone perform worse than their original counterparts, highlighting the importance of graph structure in the condensation process, as discussed in our paper. We will include this baseline in the revised final version for better comparisons.

---

### Official Review · Reviewer_oApz · 2024-07-28
**A comprehensive and valuable benchmark paper**

**Rating:** 7
**Confidence:** 4
**Correctness:** Yes
**Clarity:** Yes

**Review:**

Strengths:
1. The paper presents a thorough and systematic evaluation framework for graph condensation methods, covering multiple dimensions such as effectiveness, transferability, and efficiency.
2. The paper offers valuable insights into the current state of GC methods, such as the observation that larger condensation ratios do not necessarily lead to better performance and that existing methods struggle with transferability across different tasks and backbone architectures. These findings can guide future research and development in the field.
3. The paper identifies gaps in the current literature, such as the need for task-agnostic GC methods, better handling of complex graph structures, and improvements in efficiency and scalability. This helps in directing future research efforts toward these areas.
4. The evaluation includes a wide range of performance metrics, such as node classification accuracy, graph classification performance, time and memory consumption, and the impact of initialization strategies.

Weaknesses:
1. The paper uses five homogeneous graph datasets and two heterogeneous graph datasets for node-level tasks, and five datasets for graph-level tasks. However, the benchmark could be enhanced by including a wider variety of existing data, particularly by adding more heterogeneous graphs, including those commonly used in recommendation systems. This could broaden the usability of the proposed benchmark.
2. The benchmark includes several graph condensation methods and conducts extensive experiments on these methods. However, providing more detailed comparisons between these methods based on their performance across various datasets could be beneficial. These could help readers identify the most suitable methods for different scenarios.
3. The impact of structural properties such as heterophily and homophily on GC performance is discussed, but the exploration is limited. The benchmark could benefit from a more detailed analysis of how these properties influence different GC methods, particularly those designed for more complex graph structures.
4. The paper focuses on empirical evaluations but lacks a theoretical framework to understand the trade-offs between dataset size, information condensation, and information preservation. Establishing a theory of optimal condensation could provide deeper insights into the fundamental limits and potentials of GC methods.

**Strengths:**

see above

**Additional Feedback:**

none

**Documentation:**

The documentation is sufficient.

**Limitations:**

Yes

**Opportunities For Improvement:**

see above

**Relation To Prior Work:**

Yes

**Summary And Contributions:**

The paper introduces a comprehensive benchmark designed to evaluate the effectiveness, transferability, and efficiency of graph condensation (GC) methods. GC aims to reduce the size of large-scale graph datasets while preserving essential properties, facilitating more efficient model training and storage. The benchmark systematically assesses 12 state-of-the-art GC algorithms across 12 diverse graph datasets, covering both node-level and graph-level tasks. Key findings highlight the limitations of current methods in terms of transferability, the impact of structural properties, and the significant influence of initialization strategies on convergence speed. The authors also developed an open-source library for reproducible research, enabling easy integration of new methods and datasets. Lastly, the paper identifies critical research gaps and proposes future directions, such as task-agnostic condensation and improved handling of complex graph data, to guide ongoing advancements in the field.

---

> ### Author Rebuttal · Authors · 2024-08-17
>
> > **C3:** more detailed analysis of how structural properties influence different GC methods.
> >
>
> **A3:** Thank you for the suggestion. In response to the request for a more detailed analysis of the impact of homophily and heterophily on GC performance, we have conducted an in-depth exploration. Specifically, we calculated the correlation between the change in homophily ratio (Δ homophily) and the change in accuracy (Δ accuracy) for condensed graphs across multiple methods.
>
> **Table 1** presents the Pearson correlation coefficients between the change in homophily and the change in accuracy for different methods across three datasets. Positive correlations (e.g., 0.97 for DosCond on Cora) suggest that an increase in homophily is associated with an increase in accuracy. Negative correlations (e.g., -0.89 for DM on Cora) suggest that a decrease in homophily is associated with an increase in accuracy.
>
> **Table 1: Correlation Between Changes in Homophily and Accuracy for Each Method**
>
> | Dataset | GCDM | DM | DosCond | GCond | SGDD |
> | --- | --- | --- | --- | --- | --- |
> | Cora | 0.07 | -0.89 | 0.97 | -0.40 | -0.87 |
> | Citeseer | 0.66 | -0.95 | -0.27 | 0.91 | 0.89 |
> | Flickr | 0.64 | -0.99 | -1.00 | -0.98 | -0.52 |
>
> **Table 2** provides a detailed breakdown of the correlation between the average change in homophily and the average change in accuracy across all methods at different condensation ratios for each dataset. The original homophily values of the graphs are indicated next to the dataset names.
>
> **Table 2: Correlation at Different Condensation Ratios**
>
> | Dataset | Ratio | Correlation |
> | --- | --- | --- |
> | Cora ($\mathcal{H}$=0.81) | 1.3% | -0.84 |
> |  | 2.6% | -0.30 |
> |  | 5.2% | -0.51 |
> | Citeseer($\mathcal{H}$=0.74) | 0.90% | -0.32 |
> |  | 1.80% | -0.23 |
> |  | 3.60% | -0.50 |
> | Flickr($\mathcal{H}$=0.24) | 0.05% | 0.92 |
> |  | 0.50% | 0.26 |
> |  | 1% | 0.28 |
>
> These results indicate that for homophilous graphs like Cora and Citeseer, condensation performance improves as the condensed graph becomes more heterophilous. Conversely, for heterophilous graphs like Flickr, the opposite trend is observed.
>
> Additionally, for heterophilous graph datasets like Flickr, the performance of the condensed graph still lags behind the results of the whole graph when trained on GNNs designed for graphs with heterophily or using data augmentation, suggesting room for further improvement:
>
> |  | DEMO-Net [A] | GCN+GAugM [B] | GraphSAGE [C] |
> | --- | --- | --- | --- |
> | Flickr (Accuracy %) | 65.6 | 68.2 | 64.1 |
>
> We hope this deeper analysis addresses your concern and provides further insights into the influence of structural properties on GC methods.
>
> [A] DEMO-Net: Degree-specific Graph Neural Networks for Node and Graph Classification, Proceedings of the 25th ACM SIGKDD International Conference on Knowledge Discovery & Data Mining, 2019
>
> [B] Data Augmentation for Graph Neural Networks, Proceedings of the AAAI Conference on Artificial Intelligence, 2022
>
> [C] Inductive Representation Learning on Large Graphs, Neural Information Processing Systems,Neural Information Processing Systems, 2017
>
> > **C4:** Establishing a theory of optimal condensation could provide deeper insights into the fundamental limits and potentials of GC methods.
> >
>
> **A4:** Thank you for highlighting this important aspect. We agree that developing a theoretical framework to understand the trade-offs between dataset size, information condensation, and information preservation is crucial. We **acknowledge this in our manuscript under the Future Directions section (lines 270-272)**, where we discuss the potential for exploring whether a theory of Pareto-optimal condensation exists in the graph condensation process.
>
> We believe that approaching this from an **information-theoretic perspective** could be particularly valuable, and we plan to investigate this further in our future work.

---

> ### Author Rebuttal · Authors · 2024-08-17
>
> Dear reviewer oApz,
>
> We appreciate your detailed review. The questions you raised can indeed enhance our manuscript. Our responses are provided below.
>
> > **C1:** including a wider variety of existing data, particularly by adding more heterogeneous graphs, in recommendation systems.
> >
>
> **A1:** Indeed, including more heterogeneous datasets would enhance the usability of GC-Bench, and it's quite straightforward to extend our benchmark to other datasets. For instance, if the users have files like `label.dat`, `link.dat`, and `node.dat`, they can simply place them in the data directory. If specific indexes aren't provided, the benchmark will default to assigning 20% for validation. Additionally, for **PyG’s heterogeneous datasets** such as MovieLens or IMDB, the users can create a directory, place the files in the raw directory, or let the PyG data loader handle the download. The benchmark will automatically process them as homogeneous datasets.
>
> Regarding recommendation systems, the current graph condensation methods are primarily tailored for node and graph classification tasks, making adaptation for recommendation tasks more complex. Since transferring existing methods to recommendation tasks requires significant modifications to the models, and to ensure fairness in comparisons, recommendation datasets are not included in this benchmark at this time. We are committed to continuously updating the benchmark and repository if future methods specifically for recommendation systems become available.
>
> > **C2:** more detailed comparisons between these methods based on their performance across various datasets.
> >
>
> **A2:** Thank you for your suggestion. We appreciate the importance of providing detailed comparisons to help readers identify the most suitable methods for different scenarios.
>
> We would include specific discussions on the methods' performance in relation to different graph types and tasks. For example, trajectory matching methods like SFGC and GEOM show strong performance across a wide range of datasets, particularly on large-scale and more challenging datasets like Reddit and ogbn-arxiv. However, gradient matching methods demonstrate more stable performance in smaller, more balanced datasets like Cora and Citeseer.
>
> Furthermore, we noted that the effectiveness of these methods can vary depending on the dataset’s characteristics, such as the homophily ratio (H), which is indicated in the response to the other weakness, which can provide further insight into which methods may be more suitable depending on the structural properties of the dataset.

---

### Author Rebuttal · Authors · 2024-08-17

Dear Reviewers and ACs,

We would like to express our sincere gratitude for your time and effort in reviewing our benchmark paper. We appreciate your constructive feedback and are pleased to address your concerns. Below, we briefly summarize the common strengths and our improvements for improvement mentioned in the reviews.

### Common Strengths

1. **Thorough and Systematic Benchmark**: The benchmark covers a wide range of tasks, datasets, and algorithms, providing a thorough evaluation of graph condensation techniques. (Reviewer oApz, GyyU, 9zaP, qXuL)
2. **Important Observations and Valuable Insights**: The paper offers valuable insights into the current state of GC methods and these findings can guide future research and development in the field. (Reviewer oApz, 9zaP, qXuL)
3. **Identifies Critical Research Gaps and Future Directions:** The benchmark identifies critical research gaps and proposes future directions, such as task-agnostic condensation and improved handling of complex graph data, to guide ongoing advancements in the field. (Reviewer oApz, qXuL)
4. **Clarity and Presentation**: The paper is well-written and easy to follow. (Reviewer oApz, GyyU, 9zaP, qXuL)
5. **Promote Transparency and Reproducibility**: The provision of an open-source library to facilitate reproducible evaluation is appreciated, highlighting the practical utility of our work. (Reviewer oApz, 9zaP, qXuL)

### Our improvements according to suggestions

1. **More detailed results and comparisons:** We have added the results of Graph Transformer, the results on heterogeneous graphs, the feature-only MLP-based methods, the impact of structural properties., the performance critical difference analysis, and the impact of condensation ratio.
2. **More evaluation perspectives**: We added the comparison in the rebuttal phase including the performance on imbalance datasets, the condensation fairness, the convergence analysis, and the performance for the Neural Architecture Search (NAS) task.
3. **Clearer presentation and figures**: We have improved the presentations including the dataset format, the definition of graph condensation, and the description of Figure 3(c) and Figure 4.

We provide detailed responses to each reviewer's questions, concerns, and suggestions. Where explanations required the use of figures and tables, we have uploaded a revised one-page PDF in the corresponding response for the reviewers to consult. We would like to express our gratitude once again to all the reviewers for their valuable contributions. We believe these revisions will have a significant positive impact on our benchmark, as well as on the field of graph condensation and the broader research community.

Thank you again for your thoughtful reviews.

Sincerely,

Authors of Submission #167

---

### Decision · Program_Chairs · 2024-09-26

**Decision:**

Accept (Poster)

**Comment:**

In this submission, the authors present a graph condensation benchmark, conduct comprehensive experiments with 12 algorithms under different scenarios and diverse datasets. Useful discussions and valuable insights are also included, which will help the development of this area. All the reviewers find this submission solid.

Given these, I suggest to accept this submission.